# Epigenetic modulation of *Fgf21* in the perinatal mouse liver ameliorates diet-induced obesity in adulthood

Xunmei Yuan[1,14], Kazutaka Tsujimoto[1], Koshi Hashimoto[2], Kenichi Kawahori[1], Nozomi Hanzawa[1], Miho Hamaguchi[1,14], Takami Seki[1], Makiko Nawa[3], Tatsuya Ehara[1,4], Yohei Kitamura[4], Izuho Hatada[5], Morichika Konishi[6], Nobuyuki Itoh[7], Yoshimi Nakagawa [8,9], Hitoshi Shimano[8,9], Takako Takai-Igarashi[10], Yasutomi Kamei[11] & Yoshihiro Ogawa[1,12,13,14]

The nutritional environment to which animals are exposed in early life can lead to epigenetic changes in the genome that influence the risk of obesity in later life. Here, we demonstrate that the fibroblast growth factor-21 gene (*Fgf21*) is subject to peroxisome proliferator-activated receptor (PPAR) α–dependent DNA demethylation in the liver during the postnatal period. Reductions in *Fgf21* methylation can be enhanced via pharmacologic activation of PPARα during the suckling period. We also reveal that the DNA methylation status of *Fgf21*, once established in early life, is relatively stable and persists into adulthood. Reduced DNA methylation is associated with enhanced induction of hepatic FGF21 expression after PPARα activation, which may partly explain the attenuation of diet-induced obesity in adulthood. We propose that *Fgf21* methylation represents a form of epigenetic memory that persists into adulthood, and it may have a role in the developmental programming of obesity.

[1] Department of Molecular Endocrinology and Metabolism, Tokyo Medical and Dental University, 1-5-45 Yushima, Bunkyo-ku, Tokyo 113-8510, Japan. [2] Department of Preemptive Medicine and Metabolism, Graduate School of Medical and Dental Sciences, Tokyo Medical and Dental University, 1-5-45 Yushima, Bunkyo-ku, Tokyo 113-8510, Japan. [3] Laboratory of Cytometry and Proteome Research, Nanken-Kyoten and Research Core Center, Tokyo Medical and Dental University, 1-5-45 Yushima, Bunkyo-ku, Tokyo 113-8510, Japan. [4] Wellness and Nutrition Science Institute, Morinaga Milk Industry Co., Ltd, 5-1-83, Higashihara, Zama, Kanagawa 252-8583, Japan. [5] Laboratory of Genome Science, Biosignal Genome Resource Center, Institute for Molecular and Cellular Regulation, Gunma University, 3-39-15 Showa-machi, Maebashi, Gunma 371-8512, Japan. [6] Department of Microbial Chemistry, Kobe Pharmaceutical University, 1-19-4, Motoyama-kitamachi, Higashinada-ku, Kobe, Hyogo 658-8558, Japan. [7] Medical Innovation Center, Kyoto University Graduate School of Medicine, 53 Kawahara-cho, Shogoin, Sakyo-ku, Kyoto 606-8507, Japan. [8] Department of Internal Medicine (Metabolism and Endocrinology), Faculty of Medicine, University of Tsukuba, 1-1-1 Tennodai, Tsukuba, Ibaraki 305-8577, Japan. [9] International Institute for Integrative Sleep Medicine (WPI-IIIS), University of Tsukuba, 1-1-1 Tennodai, Tsukuba, Ibaraki 305-8577, Japan. [10] Tohoku Medical Megabank Organization, 2-1 Seiryo-machi, Aoba-ku, Sendai, Miyagi 980-8573, Japan. [11] Laboratory of Molecular Nutrition, Graduate School of Life and Environmental Sciences, Kyoto Prefectural University, 1-5 Hangi-cho, Shimogamo, Sakyo-ku, Kyoto 606-8522, Japan. [12] Department of Medicine and Bioregulatory Science, Graduate School of Medical Sciences, Kyushu University, 3-1-1 Maidashi, Higashi-ku, Fukuoka 812-8582, Japan. [13] Japan Agency for Medical Research and Development, CREST, 1-7-1 Otemachi, Chiyoda-ku, Tokyo 100-0004, Japan. [14] Present address: Department of Molecular and Cellular Metabolism, Tokyo Medical and Dental University, 1-5-45 Yushima, Bunkyo-ku, Tokyo 113-8510, Japan. Xunmei Yuan and Kazutaka Tsujimoto contributed equally to this work. Correspondence and requests for materials should be addressed to K.H. (email: khashimoto.mem@tmd.ac.jp) or to Y.O. (email: yogawa@intmed3.med.kyushu-u.ac.jp)

Nutritional experiences in early life have a long-lasting influence on the development of body weight, thus affecting the risk of obesity in later life[1,2]. For instance, malnutrition in early life as a result of poor nutrition during pregnancy and/or the lactation period may be stored onto the offspring genome as memory. It may persist into adulthood, thereby increasing the susceptibility to metabolic diseases such as obesity in later life, which has been referred to as developmental programming or the "developmental origins of health and disease (DOHaD)" hypothesis[1,3,4].

Epigenetic modifications represent a prime candidate mechanism to explain the long-lasting influence on metabolic phenotypes such as obesity[5]. Indeed, a considerable amount of evidence has recently been accumulated regarding the role of epigenetic dysregulation in human obesity[6–8].

The methylation of cytosine residues in CpG dinucleotides (i.e., cytosine followed by guanine) or DNA methylation is a major epigenetic modification known to suppress gene transcription. Cell type-specific patterns of CpG methylation are mitotically inherited, as well as highly stable in differentiated cells and tissues[9–11]. Accordingly, most epigenetic studies concerning the developmental programming of obesity focused on DNA methylation. However, whether DNA methylation status of a particular gene, when established in early life, can influence the developmental programming of obesity is currently unknown.

On the contrary, we previously reported that DNA methylation status of metabolic genes in the liver dynamically changes in early life, even during the suckling period, thus sequentially activating hepatic metabolic function to adapt to the nutritional environment[12,13].

In the previous report, we found that upon the onset of lactation after birth, milk serves as a ligand to activate the nuclear receptor peroxisome proliferator-activated receptor (PPAR)α, which is a key transcriptional regulator of hepatic lipid metabolism mediating the adaptive response to energy store[13,14,15]. PPARα activation via milk lipid ligands physiologically leads to DNA demethylation of fatty-acid β-oxidation genes in the postnatal mouse liver[13]. Given that PPARα may act as a sensor of milk lipids during the suckling period[16,17], it is likely that PPARα-dependent DNA demethylation primes the activation of the fatty-acid β-oxidation pathway in the liver, thereby contributing to the efficient production of energy from milk lipids. We also demonstrated that administration of a synthetic PPARα ligand to mouse dams during the perinatal period induces enhanced reductions in DNA methylation of fatty-acid β-oxidation genes in the liver of the offspring, suggesting that DNA methylation status of hepatic metabolism-related genes can be modulated via ligand-activated PPARα during the perinatal period. Therefore, these findings prompted us to explore whether DNA methylation status of PPARα target genes, which is modulated and established in a PPARα-dependent manner in early life, persists into adulthood, and if so, we sought to clarify how these changes influence adult metabolic phenotypes such as obesity.

Using a genome-wide analysis of DNA methylation, we identified a few PPARα target genes that underwent ligand-activated PPARα-dependent DNA demethylation during the perinatal period and whose DNA hypomethylation status persists into adulthood. Among these genes, which can be referred to as epigenetic memory genes, we focused on fibroblast growth factor 21 (FGF21), a bona fide PPARα target gene, which is a major hepatocyte-derived hormone implicated in the regulation of energy homeostasis and body weight through its effect on multiple target organs including adipose tissue[18–20].

In this study, we provide the first evidence that the PPARα-dependent Fgf21 demethylation occurs in the postnatal mouse liver. Importantly, Fgf21 methylation status can be modulated in early life, and once established it persists into adulthood and exerts long-term effects on the magnitude of gene expression response to environmental cues, which may account in part for the attenuation of diet-induced obesity.

## Results

**Genome-wide analysis of PPARα-dependent DNA demethylation.** In a previous study, we found that maternal administration of a synthetic PPARα ligand (Wy 14643, Wy) during the perinatal period induces enhanced reductions in DNA methylation of fatty-acid β-oxidation genes in the postnatal mouse liver[13]. We employed the microarray-based integrated analysis of methylation by isoschizomers (MIAMI)[21] to analyze genome-wide DNA methylation status in the livers of offspring derived from dams-administered Wy dissolved in dimethyl sulfoxide (DMSO) as vehicle (Veh) during the late gestation (from 14 to 18 days after fertilization: e14–18) and lactation periods (from 2 to 16 days after birth: D2–D16) (Fig. 1a). Accordingly, we sought to identify the genes for which DNA hypomethylation status induced via ligand-activated PPARα in the perinatal mouse liver persists into adulthood.

To clarify whether Wy was transferred to pups via the breast milk, we analyzed gastric contents, which mainly consisted of the milk derived from dams, in offspring at D16 using mass spectrometry (liquid chromatography/tandem mass spectrometry [LC/MS-MS]). As shown in Supplementary Fig. 1, LC/MS-MS detected the same precursor (mass-to-charge ratio [$m/z$], 324.06] and product peaks ($m/z$, 306.04) in both a standard sample consisting of purified Wy and milk samples from Wy-offspring (derived from Wy-treated dams), suggesting that Wy is present in the breast milk of dams (Supplementary Fig. 1).

We performed lipid composition analysis of milk using the offspring gastric contents by gas chromatography (GC). GC showed no significant difference in lipid composition of milk between Wy- and Veh-offspring (derived from Veh-treated dams), suggesting that Wy administration to dams during the lactation period did not affect milk lipid composition (Supplementary Table 1).

MIAMI analysis revealed that more genes were DNA hypomethylated in Wy-offspring relative to Veh-offspring at D16 (Fig. 1b), and were DNA hypermethylated at 14W (14W) (Fig. 1c). A correlation plot showing the differences at D16 (x-axis) vs. 14W (y-axis) indicates a weak but significant correlation between DNA methylation status at D16 and 14W (Fig. 1d). We found that 424 genes were DNA hypomethylated in Wy-offspring relative to Veh-offspring at D16, and 33 genes were DNA hypomethylated at 14W after birth (Fig. 1e). Consequently, we identified 25 genes, which were DNA hypomethylated in Wy-offspring relative to Veh-offspring both at D16 and 14W (Fig. 1e). Pathway analysis of the 25 genes yielded the PPAR signaling pathway among which 11 genes are known to be PPARα target genes[15] (Table 1).

**PPARα-dependent Fgf21 demethylation in the suckling period.** Among the 11 aforementioned PPARα target genes, we focused on FGF21, a peptide hormone which plays a critical role in regulating energy homeostasis[20]. To verify whether Fgf21 demethylation physiologically occurs in a PPARα-dependent manner, we examined Fgf21 methylation status in PPARα-deficient (PPARα-KO) and wild-type (WT) offspring via bisulfite-sequencing analysis. In silico search identified 21 CpG sites around the transcription start site (TSS) of Fgf21, with two PPAR response elements (PPRE1 and PPRE2) that are located approximately 1000 and 100 bp upstream of the TSS, respectively (Fig. 2a)[22].

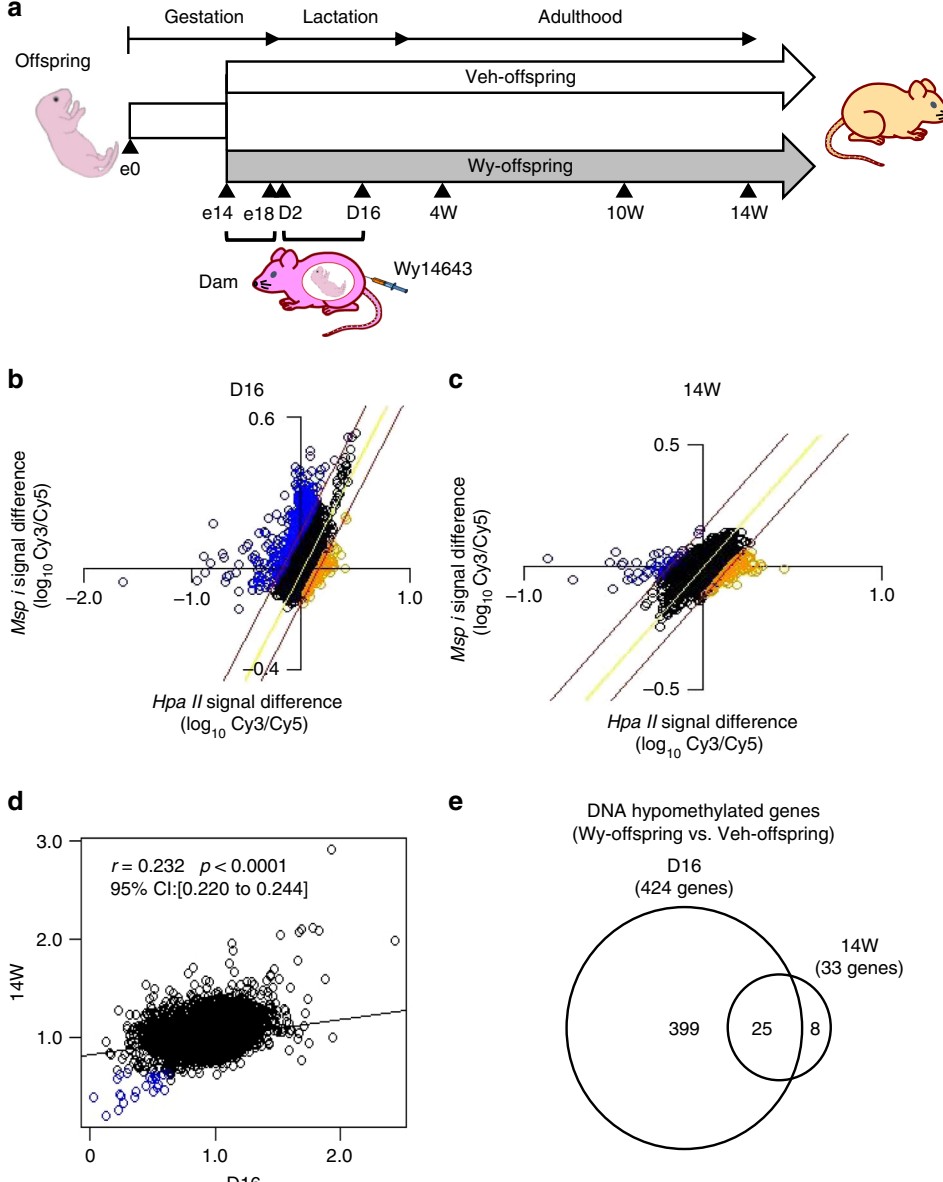

**Fig. 1** Genome-wide DNA methylation analysis of Wy-offspring. **a** Experimental protocol of genome-wide DNA methylation analysis in the liver of offspring derived from dams-administered Wy (Wy-offspring) or DMSO (Veh-offspring) during the late gestation (e14–e18) and lactation periods (D2–D16). **b**, **c** MIAMI analysis comparing vehicle (labeled with Cy3) with Wy (Cy5) at D16 (**b**) and 14W (**c**). Plots of log transformed values of *Hpa*II (methylation-sensitive, horizontal axis) and *Msp*I signal difference (methylation-insensitive, vertical axis) between samples. The values of *Hpa*II signal difference*/Msp*I signal difference judged as increased and decreased DNA methylation are above 1.4 and below 0.65, respectively. The regression line is in yellow and red lines are located $\log_{10}$ 1.4 and $\log_{10}$ 0.65 of the horizontal distance from the regression line. Orange and blue circles are judged as hypermethylated and hypomethylated, respectively. **d** A correlation plot showing the differences at 14W (*y*-axis) vs. D16 (*x*-axis). Genes, which are hypomethylated both at D16 and 14W are highlighted as blue circles. **e** Venn-diagram showing the result of MIAMI analysis. Four-hundred and twenty-four genes were DNA hypomethylated in Wy-offspring relative to that in Veh-offspring at D16 and 33 genes at 14W, respectively. Consequently, we identified 25 genes that were DNA hypomethylated in Wy-offspring relative to that in Veh-offspring both at D16 and 14W

There was no significant change in *Fgf21* methylation in the livers of WT offspring during the late fetal and early postnatal periods (e14-D2). However, there was significant *Fgf21* demethylation during the suckling period (D2–D16) (Fig. 2b, c). *Fgf21* demethylation was gradually induced from D16 to 4W, after which it remains constant from 4W to 14W (Fig. 2b, c). We also found that *Fgf21* methylation status remains unchanged in the livers of PPARα-KO offspring at all time points examined (Fig. 2b, c). Taken together, these observations suggest that PPARα-dependent *Fgf21* demethylation physiologically occurs during the suckling period, after which it persists into adulthood.

**Fgf21 demethylation is enhanced by ligand-activated PPARα.** We found that *Fgf21* demethylation is more enhanced in Wy-offspring than in Veh-offspring at D16 and 4W, after which it remained relatively constant until 14W (Fig. 2b, c). Notably, the difference in *Fgf21* methylation was maintained between Wy- and Veh-offspring at 4W (36.3% and 49.5%, respectively) and 14W (34.0% and 48.5%, respectively). These observations, taken together, suggest that *Fgf21* demethylation can be modulated and enhanced in a PPARα ligand-dependent manner in the postnatal mouse liver, and that the ligand-activated *Fgf21* demethylation established in early life persists into adulthood. In this study,

**Table 1 PPARα target genes which were DNA hypomethylated at both D16 and 14W (Wy-offspring vs. Veh-offspring)**

| Gene symbol | Gene name |
| --- | --- |
| Acot1 | Acyl-CoA thioesterase 1 |
| Aqp3 | Aquaporin 3 |
| Aqp7 | Aquaporin 7 |
| Cpt1b | Carnitine palmitoyltransferase 1b |
| Dgat1 | Diacylglycerol O-acyltransferase 1 |
| Fabp3 | Fatty-acid-binding protein 3 |
| Fgf21 | Fibroblast growth factor 21 |
| Peci | Peroxisomal 3,2-trans-enoyl-CoA isomerase |
| Plin1 | Perilipin 1 |
| Psat1 | Phosphoserine aminotransferase 1 |
| Ucp3 | Uncoupling protein 3 |

there was no significant *Fgf21* demethylation in PPARα-KO offspring, indicating that Wy-induced *Fgf21* demethylation is PPARα dependent (Supplementary Fig. 2a).

**PPARα-dependent *Fgf21* demethylation is life stage-specific.** We next examined at which life stage PPARα-dependent *Fgf21* demethylation can be ligand-activated in the mouse liver. In this study, we divided dams into two groups; one was treated with Wy only during the late gestation period (e14–e18) and the other was treated with Wy only during the lactation period (D2–D16) (Fig. 3a). We found that reductions in *Fgf21* methylation were significantly enhanced in Wy-offspring relative to Veh-offspring during the lactation but not the late gestation period. Importantly, when adult WT mice were treated directly with Wy for 2 weeks after the suckling period (4–6W), there was no significant *Fgf21* demethylation from 4W to 14W (Fig. 3b). These observations suggest that the ligand-activated PPARα-dependent *Fgf21* demethylation occurs specifically during the suckling period.

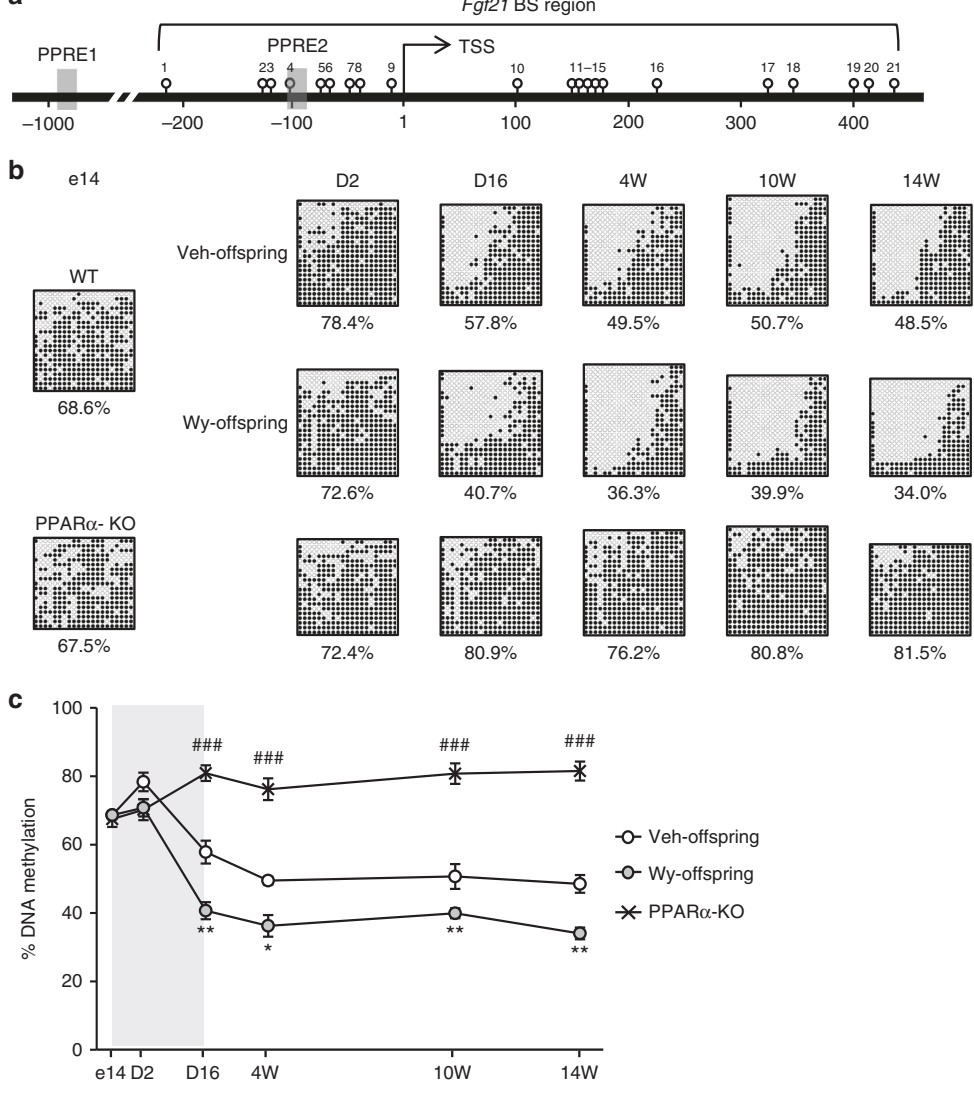

**Fig. 2** DNA methylation analysis of *Fgf21*. **a** Schematic representation of the promoter region of *Fgf21*. Open circles and gray boxes indicate CpG sites and PPAR response elements (PPREs), respectively. Bisulfite-sequencing (BS) analysis region encompassing the transcription start site is indicated. **b** Bisulfite-sequencing analysis of *Fgf21* in Wy- and DMSO (Veh)-treated offspring and PPARα-KO mice. Closed and open circles indicate methylated and unmethylated CpGs, respectively. Representative data of three independent experiments are shown. **c** Graphic presentation of statistical analysis of the bisulfite-sequencing data. The gray-shaded box indicates the period of maternal administration of Wy or Veh, (n = 3–6 at each time point). Statistics by one-way ANOVA with Tukey's multiple comparison test. Data are expressed as mean ± SEM. *P < 0.05; **P < 0.01; ###P < 0.001 vs. Veh-offspring

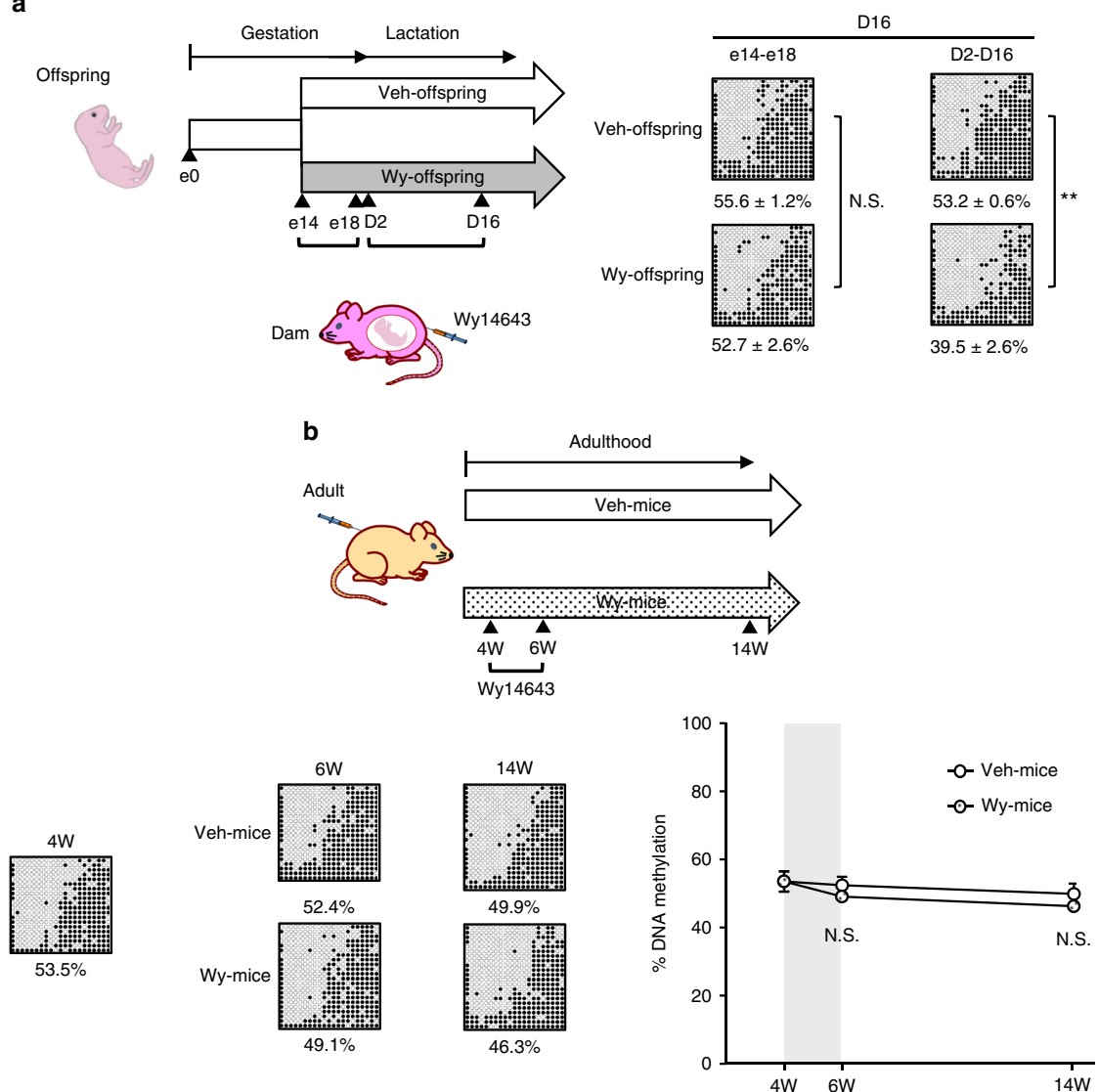

**Fig. 3** Life stage-specific DNA demethylation of *Fgf21*. **a** Experimental protocol of maternal administration of Wy or DMSO (Veh) during the late gestation (e14–e18) and lactation periods (D2–D16) (left). Bisulfite-sequencing analysis of the offspring on D16 (right) (*n* = 3–4 for each group). Statistics by unpaired Student's *t*-test. Data (% DNA methylation) are expressed as mean ± SEM. **P < 0.01; N.S.: not significant vs. Veh-offspring. **b** Experimental protocol (top) and bisulfite-sequencing analysis (bottom) in mice treated with Wy or Vehicle from 4W to 6W (Wy- and Veh-mice, respectively). Bisulfite-sequencing analysis of *Fgf21* in Wy- and Veh-mice (bottom left). Representative data of three independent experiments are shown. Graphic presentation of statistical analysis of the bisulfite-sequencing data (bottom right). The gray-shaded box indicates the period of Wy or Veh administration. (*n* = 4–5 per group, statistics by one-way ANOVA with Tukey's multiple comparison test). Data are expressed as mean ± SEM. **P < 0.01; N.S., not significant vs. Veh-mice

**Milk lipid components responsible for *Fgf21* demethylation.** The lipid composition analysis of milk using the gastric contents of offspring revealed that palmitic acid, oleic acid and linoleic acid, which have been proposed as natural endogenous ligands for PPARα[23,24], are the major components of milk lipids (Supplementary Table 1).

Because fatty acids in diet influence the lipid component of breast milk of dams[25], we administered fat-free diet (0% energy as fat) and control diet (10% energy as fat) to dams during late gestation and lactation period and analyzed *Fgf21* methylation in the liver of offspring (Supplementary Fig. 3). The lipid composition analysis of milk using the gastric contents of offspring derived from fat-free diet-fed dams revealed that linoleic acid and α-linolenic acid levels were markedly reduced relative to offspring derived from control diet-fed dams; notable,

that eicosapentaenoic acid (EPA) was not detected in the milk of fat-free diet-fed dams (Supplementary Table 2).

However, *Fgf21* methylation status of the offspring derived from fat-free diet-fed dams was mainly unchanged relative to that derived from control diet-fed dams both at D16 and 4W (Supplementary Fig. 3). Because the ratios of palmitic acid, oleic acid, arachidonic acid (ARA), and docosahexaenoic acid (DHA), in milk, which are known to be ligands for PPARα[23,24], were comparable between the offspring derived from fed fat-free and control diet dams, it is likely that they are responsible for physiological *Fgf21* demethylation. Moreover, as the physiological *Fgf21* demethylation was induced without EPA in the milk components, we speculated that EPA may not be related to *Fgf21* demethylation (Supplementary Table 2).

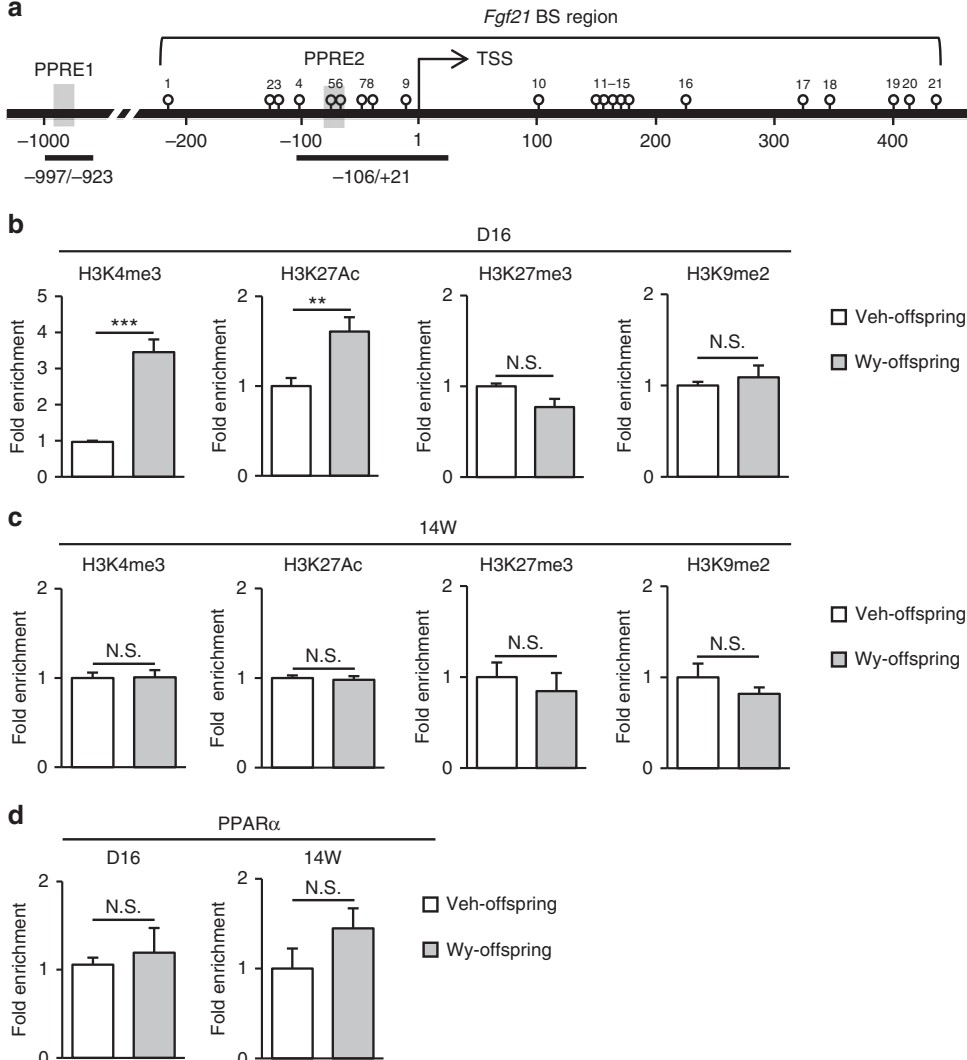

**Fig. 4** Histone modifications of *Fgf21* in Wy-offspring. **a** Schematic representation of the promoter region of *Fgf21*. Open circles and gray boxes indicate CpG sites and PPAR response elements (PPREs), respectively. BS, bisulfite sequencing. **b**, **c** ChIP assays of histone marks in Wy- and Veh-offspring at D16 (**b**) and 14W (**c**), with the indicated antibodies. Primers amplifying the region of −106 to +21 bp were used for ChIP-qPCR analysis ($n = 5$–6 per group). **d** ChIP assays of the recruitment of PPARα to PPRE1. Primers amplifying the region of −997 to −923 bp were used for ChIP-qPCR analysis ($n = 4$–8 per group). Statistics by unpaired Student's *t*-test. Data are expressed as mean ± SEM. **$P < 0.01$; ***$P < 0.001$; N.S., not significant vs. Veh-offspring

**Reductions in *FGF21* methylation in the fetal human liver**. We also examined human *FGF21* methylation status in fetal and adult livers. The locations of PPRE (PPRE1 but not PPRE2 in the mouse sequence), TSS, and potential CpG sites for DNA methylation are conserved between mice and humans (Supplementary Fig. 4a). In this study, *FGF21* methylation was markedly reduced in the adult liver than in the fetal liver (Supplementary Fig. 4b), suggesting a common mechanism of developmental programming in mammals.

**Histone modification of *Fgf21* in the postnatal mouse liver**. Using chromatin immunoprecipitation (ChIP) assays (Fig. 4a), we found that H3K4me3 and H3K27Ac, two transcriptionally active histone marks, are enriched in *Fgf21* promoter in Wy-offspring relative to the findings in Veh-offspring at D16, when the levels of the repressive histone marks, H3K9me2 and H3K27me3 were roughly comparable between Wy- and Veh-offspring (Fig. 4b). At 14W, however, there was no significant difference in the levels of active and repressive histone marks in

*Fgf21* promoter (Fig. 4c). On the other hand, the recruitment of PPARα was roughly equivalent between Wy- and Veh-offspring both at D16 and 14W (Fig. 4d).

**TET2 may be related to PPARα-dependent *Fgf21* demethylation**. We evaluated mRNA expressions for epigenetic modifiers such as Ten-eleven translocation (TET) enzymes and DNA methyltranferases (DNMTs)[11,26]. Postnatal ontogenic gene expression of TET enzymes showed that *Tet1* mRNA levels were decreased in a time-dependent manner after birth, whereas both *Tet2* and *Tet3* mRNA levels were increased with a peak at D16 and declined thereafter. On the other hand, both *Dnmt3a* and *Dnmt3b* mRNA levels were gradually decreased toward adulthood (Fig. 5a).

We performed ChIP assays for these epigenetic modifiers at D16; TET2 but not TET1 or TET3 was recruited abundantly to *Fgf21* promoter in Wy-offspring than in Veh-offspring (Fig. 5b). On the other hand, there was no significant difference in the recruitment of DNMT3a and DNMT3b to *Fgf21* promoter

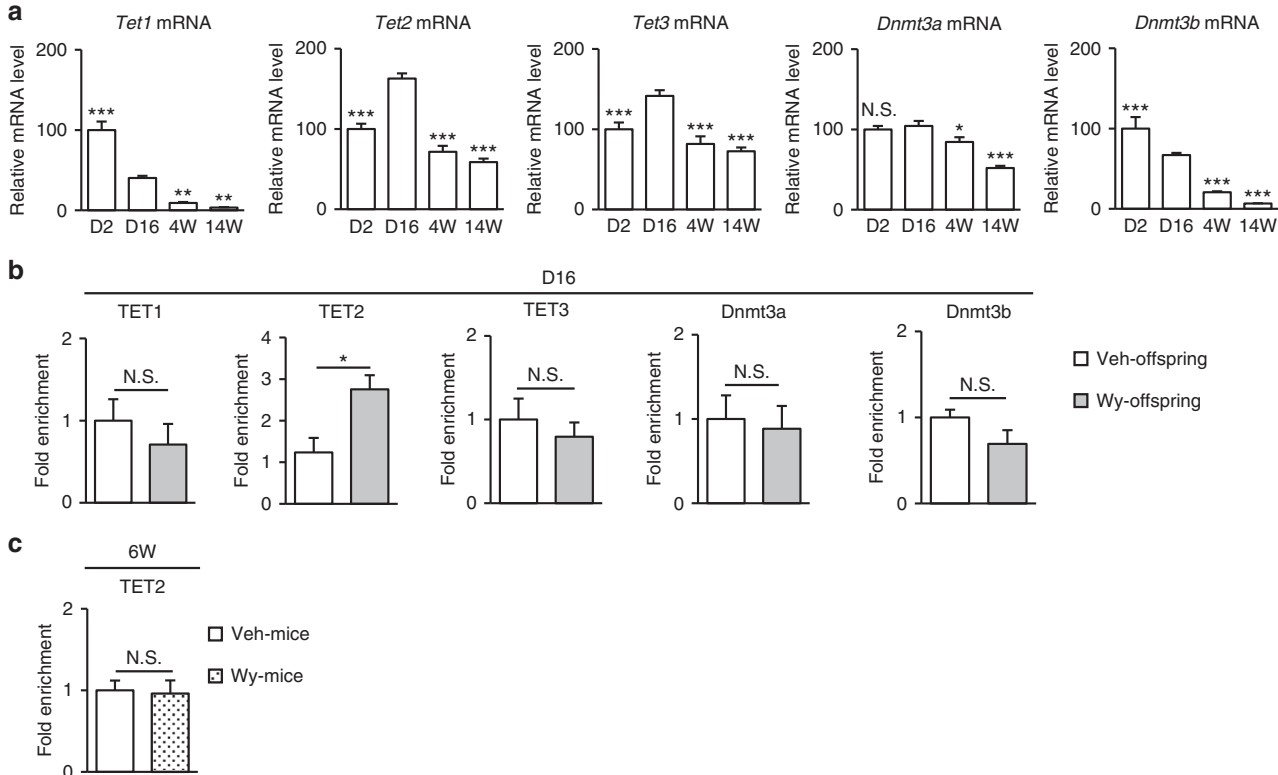

**Fig. 5** Enhanced recruitment of TET2 to *Fgf21* promoter in Wy-offspring at D16. **a** Hepatic *Tet1, Tet2, Tet3, Dnmt3a*, and *Dnmt3b* mRNA expression in mice from D2 to 14W ($n = 4$–8 per group, statistics by one-way ANOVA with Tukey's multiple comparison test). Data are expressed as mean ± SEM. *$P < 0.05$; **$P < 0.01$; ***$P < 0.001$; N.S., not significant vs. D16. **b** ChIP assays at D16 with the indicated antibodies. Primers amplifying the region of −106 to +21 bp were used for ChIP-qPCR analysis (Fig. 4a) ($n = 4$–8 per group, statistics by unpaired Student's *t*-test). Data are expressed as mean ± SEM. *$P < 0.05$; N.S., not significant vs. Veh-offspring. **c** ChIP assays at 6W for TET2 in mice treated with Wy (Wy-mice) or DMSO (Veh-mice) from 4W to 6W (Fig. 3b) ($n = 4$ per group, statistics by unpaired Student's *t*-test). Data are expressed as mean ± SEM. N.S., not significant vs. Veh-mice

between Wy- and Veh-offspring at D16 (Fig. 5b). In this study, adult WT mice, when treated directly with Veh (Veh-mice) or Wy (Wy-mice) for 2 weeks after the suckling period (4–6W), showed no significant difference in *Fgf21* methylation status between 4W and 6W (Fig. 3b). At 6W, there was no significant difference in the recruitment of TET2 to *Fgf21* promoter between Veh- and Wy-mice (Fig. 5c).

**Fgf21 methylation status determines gene expression response**. We examined hepatic *Fgf21* mRNA expression and serum FGF21 concentrations in Wy- and Veh-offspring. We detected substantial amounts of hepatic *Fgf21* mRNA and serum FGF21 protein in Veh-offspring on D2 and D16, possibly as a result of neonatal starvation and/or in response to milk lipid intake[16], whereas their levels were relatively low at 4W and 14W (Fig. 6a, b). This is consistent with previous reports that hepatic *Fgf21* mRNA expression and serum FGF21 concentrations are rapidly increased upon birth, after which they remain elevated during the suckling period and gradually decline in adulthood[16,18].

We next examined whether *Fgf21* methylation status affects *Fgf21* mRNA expression in adulthood upon transient PPARα activation. There was no significant difference in steady-state hepatic *Fgf21* mRNA levels and serum FGF21 concentrations between Wy- and Veh-offspring (Fig. 6c, d: left). However, upon PPARα activation by a single Wy injection, *Fgf21* mRNA expression was significantly increased in Wy-offspring than in Veh-offspring (Fig. 6c, d: left). There was a negative correlation

between the degree of DNA methylation (% DNA methylation) and the induction of gene expression (Fig. 6c, d: right).

Because FGF21 expression is known to increase during fasting as a result of PPARα activation[18,22,27–29], we examined the effect of fasting on serum FGF21 concentrations. In response to 24-h fasting, serum FGF21 concentrations were significantly increased in Wy-offspring than in Veh-offspring at 14W (Fig. 6e). In this study, there was no significant difference in serum non-esterified fatty-acid (NEFA) concentrations after 24-h fasting between Wy- and Veh-offspring (Fig. 6f), suggesting that the increased induction of serum FGF21 is due to increased responsiveness to rather than increased levels of endogenous ligands for PPARα.

As shown in Fig. 3b, Wy- and Veh-mice displayed no significant change in DNA methylation status of *Fgf21* at 14W. Upon a single Wy injection at 14W, serum FGF21 concentrations were similarly increased in both Wy- and Veh-mice (Supplementary Fig. 5).

ChIP assays revealed that RNA polymerase II (Pol II) was recruited abundantly to *Fgf21* promoter of Wy-offspring relative to Veh-offspring at D16 (Fig. 6g), suggesting active transcription of *Fgf21* in Wy-offspring relative to Veh-offspring. Upon a single Wy injection at 14W, Pol II was recruited abundantly to *Fgf21* promoter of Wy-offspring relative to Veh-offspring (Fig. 6h). This data suggest that a modest difference in DNA methylation status can affect transcriptional activity.

We evaluated DNA methylation ratios of each CpG site in *Fgf21* promoter at 14W (Supplementary Fig. 6a) and found that CpG sites located downstream of the TSS (the CpG site number:

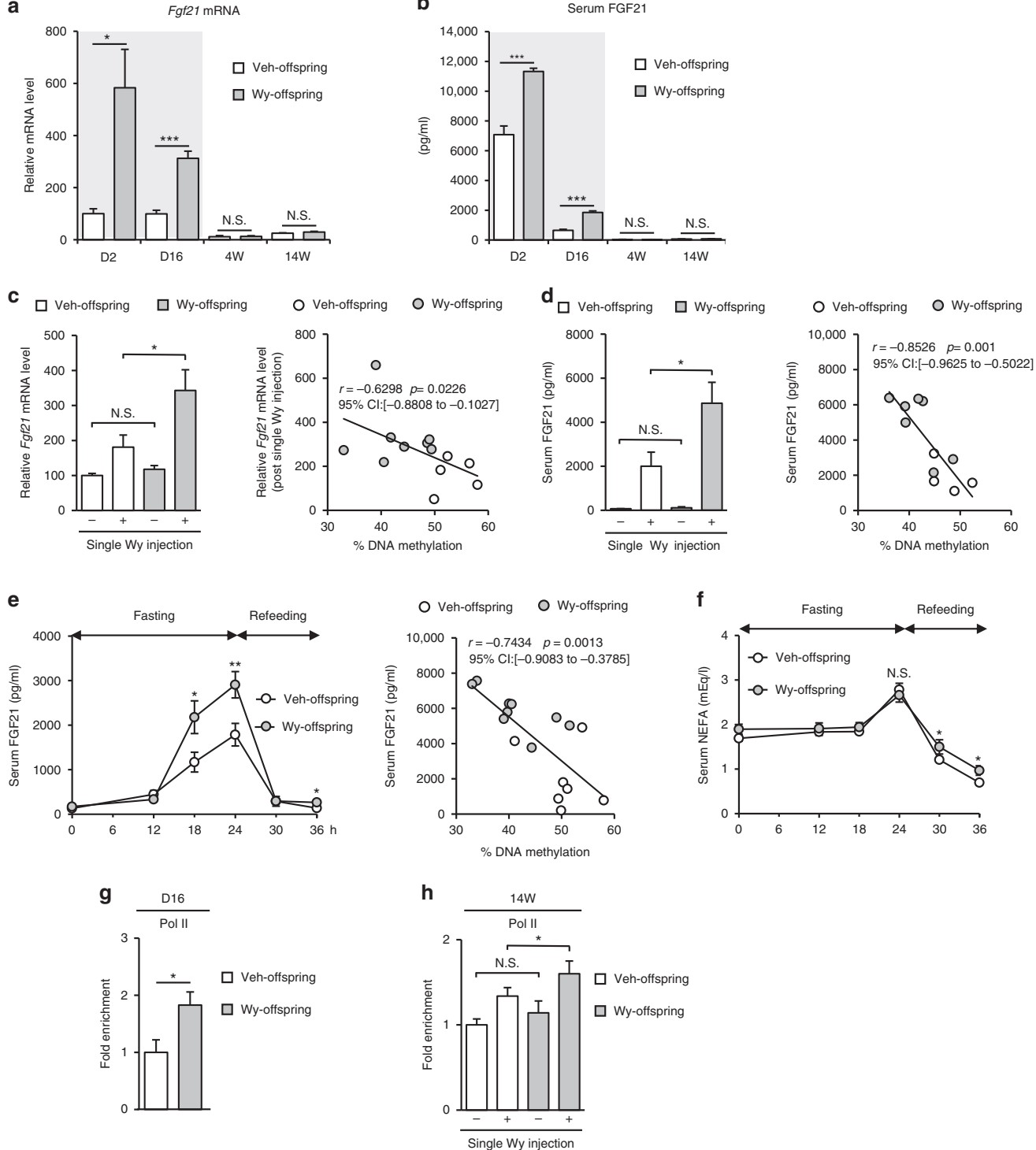

**Fig. 6** Correlation between DNA methylation status of *Fgf21* and FGF21 expression. **a, b** Hepatic *Fgf21* mRNA expression and serum FGF21 concentrations in Wy- and DMSO (Veh)-treated offspring from D2 to 14W. The gray-shaded box indicates the period of maternal administration of Wy or Veh (*n* = 8 per group). **c** Hepatic *Fgf21* mRNA expression after a single Wy injection in Wy- and Veh-offspring at 14W (left, *n* = 5–8 per group). Correlation between *Fgf21* mRNA expression and DNA methylation (right, *n* = 13). **d** Serum FGF21 concentrations after a single Wy injection in Wy- and Veh-offspring at 14W (left, *n* = 4–7 per group). Correlation between serum FGF21 concentrations and DNA methylation (right, *n* = 11). **e** Serum FGF21 concentrations during fasting and refeeding (left, *n* = 7–9 per group). Correlation between serum FGF21 concentrations after a 24-h fast and DNA methylation (right, *n* = 16). **f** Serum NEFA concentrations during fasting and refeeding (*n* = 7–9 per group). **g, h** ChIP assays for Pol II in Wy- and Veh-offspring at D16 (**g**) and with a single DMSO (Veh) or Wy injection at 14W (**h**). Primers amplifying the region of −106 to +21 bp were used for Pol II (Fig. 4a) (*n* = 7–8 per group). Statistics by unpaired Student's *t*-test (**a, b, c–e** left, **f, g, h**) or Spearman's rank correlation coefficient (**c–e** right). The *r*- and *p*-values are indicated on the graph. Data are expressed as mean ± SEM. *$P < 0.05$; **$P < 0.01$; ***$P < 0.001$; N.S., not significant vs. Veh-offspring

#10–21) are sensitive to PPARα-dependent DNA demethylation. The CpG sites but not those located upstream of the TSS were correlated to the induction of *Fgf21* expression upon the single Wy injection (Supplementary Fig. 6b, c).

**Wy-offspring exhibited less weight gain during HFD feeding.** We examined metabolic phenotypes in Wy- and Veh-offspring during 10 weeks of high-fat diet (HFD) or normal chow diet (NCD) feeding (4–14W) (Fig. 7a and Supplementary Fig. 7a).

Animals were classified as Veh-offspring fed NCD (Veh-NCD) or HFD (Veh-HFD) and Wy-offspring fed NCD (Wy-NCD) or fed HFD (Wy-HFD). In this study, we found no significant difference in body weight and food intake between Wy-NCD and Veh-NCD (Supplementary Fig. 7b, c). The weights of inguinal white adipose tissue (iWAT), epididymal WAT (eWAT), and brown adipose tissue (BAT) were roughly comparable between the Wy-NCD and Veh-NCD groups at 14W (Supplementary Fig. 7d). There was no significant difference in serum triglyceride (TG) and total cholesterol (T-Chol) concentrations, or glucose tolerance

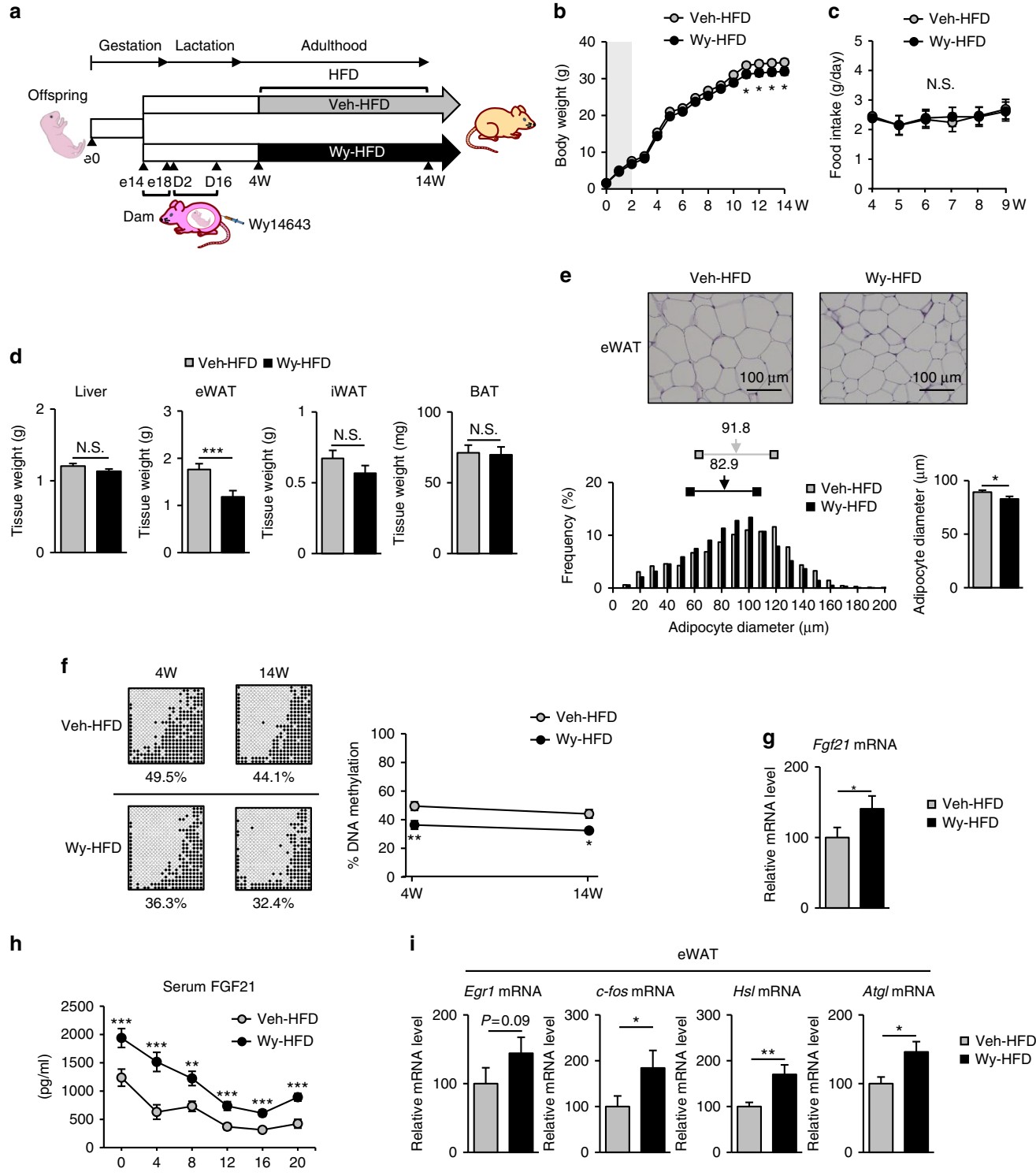

(Supplementary Fig. 7e, f). Serum FGF21 concentrations in both Wy-NCD and Veh-NCD were relatively low, with no appreciable difference (Supplementary Fig. 7g).

After 10 weeks of HFD feeding, Wy-HFD gained less weight than Veh-HFD (Fig. 7b) even though food intake was similar (Fig. 7c). The weight of eWAT was significantly lower in Wy-HFD than in Veh-HFD (Fig. 7d). Histologically, adipocyte cell size in eWAT appeared to be reduced in Wy-HFD relative to that in Veh-HFD (Fig. 7e).

We found no significant difference in serum TG, T-Chol, NEFA, and adiponectin concentrations between Wy-HFD and Veh-HFD (Supplementary Fig. 8a). The area under the curve of the glucose tolerance test (GTT) was significantly reduced in Wy-HFD relative to that in Veh-HFD (Supplementary Fig. 8b).

**Enhanced FGF21 expression in Wy-offspring during HFD feeding.** Bisulfite-sequencing analysis revealed that although *Fgf21* demethylation is marginally induced in both Wy-HFD and Veh-HFD, the difference in *Fgf21* methylation status between Wy-HFD and Veh-HFD remains constant even after HFD feeding (Fig. 7f).

Hepatic *Fgf21* mRNA expression was significantly higher in Wy-HFD than in Veh-HFD at 14W (Fig. 7g). In this study, serum FGF21 concentrations in Wy-HFD ranged 800–2000 pg/mL, and the circadian variation in serum FGF21 concentrations was approximately twofold higher in Wy-HFD than in Veh-HFD at all time points examined (Fig. 7h). Consistently, the mRNA expression of *Egr1* and *c-fos*, which are located downstream of *Fgf21*[30], and that of *Hsl* and *Atgl*, which are lipolytic enzymes, were increased in eWAT from Wy-HFD relative to those in Veh-HFD (Fig. 7i). The mRNA expression of these genes was roughly equivalent between Wy-HFD and Veh-HFD both in iWAT and BAT (Supplementary Fig. 8c, d), suggesting the selective response of fat depots in response to FGF21.

**Similar phenotypes of rhFGF21- and Wy-treated HFD-fed mice.** To explore whether increased FGF21 production contributes to the metabolic phenotypes in Wy-HFD, we examined the effect of recombinant human (rh) FGF21 administration in adult WT mice fed an HFD for 4 weeks (4–8W) (Supplementary Fig. 9a). We optimized the rhFGF21 dose to ensure that the increase in serum FGF21 concentrations in FGF21-treated mice relative to that in saline-treated mice is roughly comparable to that in Wy-HFD relative to the findings in Veh-HFD (Fig. 7h). In this study, we found that serum FGF21 concentrations were approximately 1300 and 400 pg/mL in FGF21- and saline-treated mice, respectively (Supplementary Fig. 9b).

The FGF21-treated HFD-fed mice displayed significantly lower body weight and eWAT weight than saline-treated HFD-fed mice (Supplementary Fig. 9c, d). Histologically, adipocyte cell size in eWAT from FGF21-treated HFD-fed mice appeared to be smaller than that from saline-treated HFD-fed mice (Supplementary Fig. 9e). *Egr1*, *c-fos*, *Hsl*, and *Atgl* mRNA expression was higher in FGF21-treated HFD-fed mice than in saline-treated HFD-fed mice (Supplementary Fig. 9f).

**Metabolic phenotypes of Wy-HFD were alleviated in FGF21-KO.** We obtained FGF21-deficient (KO) mice[31] and induced DNA demethylation of all the PPARα target genes except *Fgf21* by maternal administration of Wy (Fig. 8a). We examined the metabolic phenotype of the offspring derived from FGF21-KO dams treated with Wy and Veh during 10 weeks of HFD feeding (4–14W) (Veh-FGF21-KO and Wy-FGF21-KO, respectively) (Fig. 8a). As shown in Fig. 8b, we found no significant difference in body weight between Veh-FGF21-KO and Wy-FGF21-KO. The weights of iWAT, eWAT, and BAT were roughly comparable between the Veh-FGF21-KO and Wy-FGF21-KO at 14W (Fig. 8c). Serum FGF21 was not detected in both groups, proving systemic FGF21 deficiency (Fig. 8d). Histologically, adipocyte cell size in eWAT appeared to be comparable between the Veh-FGF21-KO and Wy-FGF21-KO groups at 14W (Fig. 8e). Consistently, the mRNA expression of *Egr1*, *c-fos*, *Hsl*, and *Atgl* showed no significant difference between the two groups (Fig. 8f). These observations are consistent with the notion that *Fgf21* plays a major role in the metabolic phenotypes of Wy-HFD (Fig. 7).

**Discussion**

This study represents the first detailed analysis of DNA methylation status of a particular gene throughout life. Hence, we demonstrated that *Fgf21* methylation status can be modulated in a PPARα-dependent manner and that once established in early life, the status persists into adulthood. Given that PPARα may act as a sensor of milk lipids during the suckling period[16,17], it is likely that the suckling period provides a critical time window for PPARα-dependent *Fgf21* demethylation in response to the maternal environment.

The detailed mechanism underlying PPARα-dependent DNA demethylation of its target genes remains to be elucidated. A couple of studies proposed that transcriptional factors such as PPARγ and aryl hydrocarbon receptor (Ahr) induce DNA demethylation by TET enzymes[32,33]. Given that TET2 but not TET1 or TET3 were significantly more abundantly recruited to *Fgf21* promoter in Wy-offspring than in Veh-offspring at D16, TET2, an eraser of DNA methylation, may be involved in *Fgf21* demethylation during the suckling period. Furthermore, under PPARα deficiency, the recruitment of TET2 to *Fgf21* promoter was roughly equivalent between Wy- and Veh-offspring (Supplementary Fig. 2b), thereby indicating the potential interaction between TET2 and PPARα. In adulthood, we found no differences in the recruitment of TET2 between the mice treated with vehicle and those treated with Wy (Fig. 5c), which may explain that PPARα-dependent *Fgf21* demethylation occurs only during

**Fig. 7** Metabolic phenotypes of Wy-offspring during high-fat diet (HFD) feeding. **a** Experimental protocol of Veh-offspring and Wy-offspring fed HFD diet, which are referred to as Veh-HFD and Wy-HFD, respectively. The gray-shaded box indicates the period of maternal administration of Wy or Veh. **b, c** Body weight changes (**b**) and total food intake during HFD feeding (**c**) (*n* = 11 per group, statistics by two-way ANOVA with repeated measures). **d** Tissue weight of Wy- and DMSO (Veh)-treated HFD mice at 14W (*n* = 11 per group). **e** Hematoxylin and eosin (HE) staining (top, representative image of ten individuals per group) and quantification of adipocyte diameter (bottom) of eWAT. Histograms of adipocyte diameter (bottom left). Horizontal lines with bilateral squares indicate interquartile range (IQR). Arrows indicate the median values (numbers above the horizontal lines) of Veh-HFD and Wy-HFD. Statistical analysis (bottom right) of mean adipocyte diameters are shown. Scale bar = 100 μm (*n* = 10 per group). **f** Bisulfite-sequencing analysis (left, representative data of three independent experiments) and graphical presentation of statistical analysis (right, *n* = 4–5 per group) of *Fgf21* in Wy-HFD and Veh-HFD at 4W and 14W. **g** Hepatic *Fgf21* mRNA expression in Wy-HFD and Veh-HFD at 14W. (*n* = 10 per group). **h** Circadian variation of serum FGF21 concentrations. ZT, zeitgeber time (*n* = 11 per group). **i** Relative mRNA expression of *Egr1*, *c-fos*, *Hsl*, and *Atgl* in eWAT (*n* = 10 per group). Statistics by unpaired Student's *t*-test otherwise indicated. Data are expressed as mean ± SEM. \*P < 0.05; \*\*P < 0.01; \*\*\*P < 0.001; N.S., not significant vs. Veh-HFD

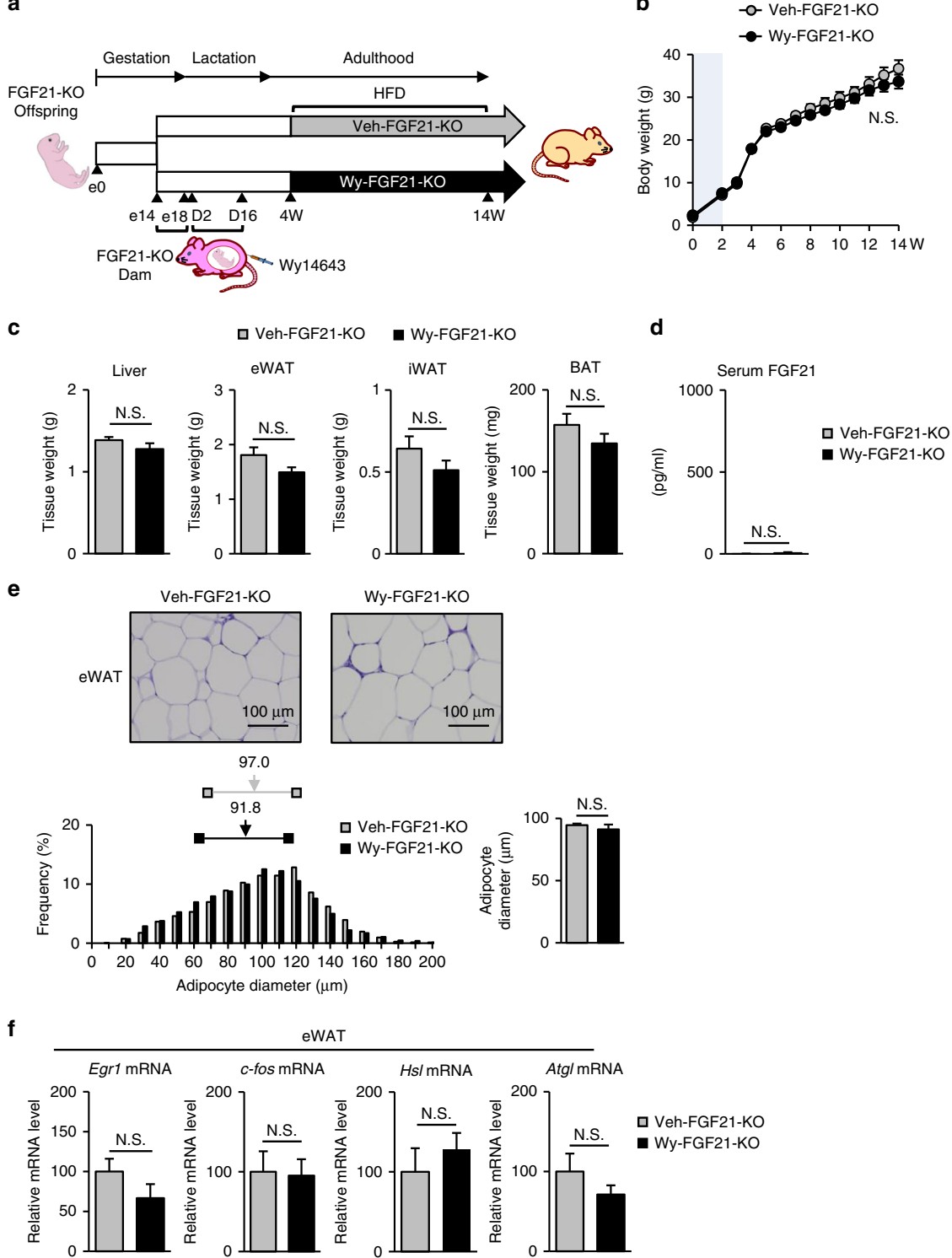

**Fig. 8** Analysis of FGF21-deficient (KO) mice. **a** Experimental protocol of maternal administration of Wy or DMSO (Veh) to FGF21-KO dams during the late gestation (e14–e18) and lactation periods (D2–D16). Offspring derived from Wy- or Veh-administered dams are referred to as Wy-FGF21-KO and Veh-FGF21-KO, respectively. These offspring were treated with HFD diet for 10 weeks in adulthood (from 4W to 14W). **b** Body weight changes during HFD feeding (n = 7–9 per group, statistics by two-way ANOVA with repeated measures). **c** Tissue weight of Wy- and Veh-FGF21-KO at 14W. **d** Serum FGF21 levels of Wy- and Veh-FGF21-KO at 14W. **e** Hematoxylin and eosin (HE) staining (top, representative image of 7–9 individuals per group) and quantification of adipocyte diameter (bottom) of eWAT. Histograms of adipocyte diameter (bottom left). Horizontal lines with bilateral squares indicate interquartile range (IQR). Arrows indicate the median values (numbers above the horizontal lines) of Veh-FGF21-KO and Wy-FGF21-KO. Statistical analysis (bottom right) of mean adipocyte diameters ± SEM are shown. Scale bar = 100 μm. **f** Relative mRNA expression of *Egr1*, *c-fos*, *Hsl*, and *Atgl* in eWAT, n = 7–9 per group. Statistics by unpaired Student's *t*-test otherwise indicated. Data are expressed as mean ± SEM. N.S., not significant vs. Veh-FGF21-KO

the suckling period. On the other hand, DNMT3a and 3b, which are de novo DNMTs; writers of DNA methylation[34] may not be related to PPARα-dependent *Fgf21* demethylation.

It is of great physiological significance to identify a milk lipid component that acts as a PPARα agonist and might thus mediate *Fgf21* demethylation.

Through milk lipid composition analysis of veh- and Wy-treated and fat-free diet-fed dams, we speculated that the fatty acids such as palmitic acid, oleic acid, ARA, and DHA, forming a coordination ligand complex for PPARα, would mediate PPARα-dependent *Fgf21* demethylation.

DNA methylation ratios of CpG sites downstream of TSS, in gene body, were significantly different between Veh- and Wy-offspring, suggesting that enhanced DNA demethylation via ligand-activated PPARα could specifically occur at these CpG sites. Since DNA methylation ratios of these CpG sites were negatively correlated to the induction of gene expression, DNA hypomethylation status at these CpG sites could determine the magnitude of the gene expression response to environmental cues. So far, possible functions of gene body or intragenic DNA methylation has not yet been fully elucidated[35]. It was reported that intragenic DNA methylation in mammalian cells initiates formation of a chromatin structure that reduces the efficacy of Pol II elongation, thereby repressing the gene expression[36], which is compatible to our data.

Bisulfite-sequencing analysis also revealed that *Fgf21* methylation status was not monotonous, suggesting cellular heterogeneity in DNA methylation status. Furthermore, because the liver consists of various types of cells such as hepatocytes, stellate cells, Kupffer cells, and sinusoidal endothelial cells[37], it may be ideal to collect hepatocytes for bisulfite sequencing.

When reductions in *Fgf21* methylation is more enhanced in Wy-offspring than in Veh-offspring, hepatic induction of *Fgf21* is exaggerated in Wy-offspring upon transient PPARα activation, during fasting and HFD feeding. Indeed, it has been known that subtle but significant differences in DNA methylation would induce substantial differences in gene expression[38]. Thus, it is likely that the degree of *Fgf21* methylation determines the magnitude of the gene expression response to environmental cues. This is consistent with a previous report that DNA demethylation of some mammary gland genes is induced during pregnancy, and once established during the first pregnancy, the DNA demethylation status remains in subsequent cycles to prime the activation of gene expression networks that promote mammary gland function[39].

It is noteworthy that the hepatic induction of FGF21 expression during fasting or HFD feeding is largely PPARα-dependent[18,20,40]. In the liver, FGF21 expression is induced by protein insufficiency, secondary to amino-acid deprivation, an effect that is downstream of activating transcription factor 4 (ATF4). Carbohydrate response element–binding protein (ChREBP), a transcription factor that regulates de novo lipogenesis in response to carbohydrate load, also increases hepatic FGF21 expression[20]. Whether PPARα-dependent *Fgf21* demethylation determines the response to these PPARα-independent environmental cues awaits further investigation.

Histone modification is another candidate mediator of epigenetic memory[11,41,42]. Indeed, a previous report illustrated that transient neonatal activation of a nuclear receptor, namely the constitutive androstane receptor, leads to permanent histone modification and induces the expression of its target genes, which confers life-long changes in hepatic drug metabolism[43]. In this study, we did not observe a significant difference of histone modification at 14W, although active marks were more enriched in Wy-offspring than in Veh-offspring on D16. However, given the targeted nature of ChIP assays, and in light of the trends

observed in several histone marks at the targeted loci, it should be noted that the data do not completely rule out the contribution of histone-related gene silencing/desilencing mechanisms to the epigenetic memory.

Importantly, the hepatic induction of FGF21 expression is more enhanced in Wy-HFD than in Veh-HFD during 10 weeks of HFD feeding, when Wy-HFD displays reduced body weight and adipose tissue mass with increased expression of lipolytic genes in eWAT relative to the findings in Veh-HFD. Even though statistically significant, the differences in body and eWAT weight between Wy-HFD and Veh-HFD were relatively small. We speculated that alternative protocols for HFD feeding, for example, HFD feeding starts after 10W, might enhance the difference.

Given that chronic rhFGF21 administration, which achieves serum FGF21 concentrations roughly equivalent to those found in Wy-HFD, reproduces some of the improved metabolic phenotypes of Wy-offspring, it is conceivable that *Fgf21*, when upregulated due to increased DNA demethylation, contributes to the metabolic phenotypes of Wy-HFD.

It is noteworthy that Wy-NCD does not differ significantly from Veh-NCD regarding steady-state *Fgf21* mRNA levels and serum FGF21 concentrations. Consistently, there was no

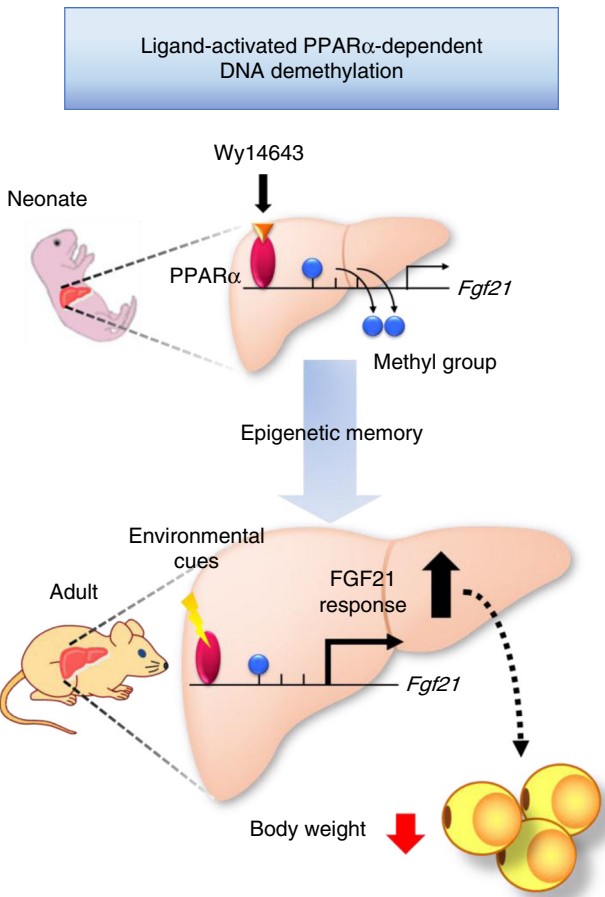

**Fig. 9** Schematic representation of epigenetic memory of *Fgf21*. Ligand-activated PPARα induces DNA demethylation of *Fgf21* in the postnatal mouse liver; DNA methylation status of *Fgf21*, once established in early life, persists into adulthood, as an epigenetic memory. In adulthood, DNA methylation status determines the magnitude of *Fgf21* expression response to environmental cues. This may lead to the reduction of body weight and adipose tissue mass during HFD feeding in adulthood

appreciable difference in metabolic phenotypes between Wy-NCD and Veh-NCD, although Wy-NCD displayed marked DNA demethylation of *Fgf21* relative to that in Veh-NCD. Because FGF21 expression is relatively low in adulthood[16,18], altered epigenetic regulation of FGF21 expression via DNA methylation may only be detected by pharmacologic activation of PPARα, during fasting, or during HFD feeding. Otherwise, *Fgf21* methylation status, once established in early life, may represent an adaptation mechanism against an additional environmental challenge in adulthood.

In addition to *Fgf21*, it is important to identify other epigenetic memory genes among the PPARα target genes. We investigated transcriptional factor binding motifs in the promoter regions of the 11 PPARα target genes, which were identified as "epigenetic memory genes" by MIAMI analysis and found no common consensus motifs other than PPRE. Since the microarray in MIAMI analysis only contains 60-mer-portions of *Hpa*II fragments located in promoter regions[21], it does not necessarily cover all the CpG sites in genes. Therefore, the results of MIAMI analysis may reflect DNA methylation status in a limited region of genes, which may sometimes make a difference with that of bisulfite sequencing to target a long region containing the CpG sites. In this study, we performed bisulfite sequencing for *Ucp3* and found that *Ucp3* methylation status at D16 does not change at 14W, and that reductions in *Ucp3* methylation are enhanced in Wy-offspring relative to Veh-offspring at D16, showing that the difference in DNA methylation between Wy- and Veh-offspring was maintained at 14W, which was similar to that with *Fgf21* (Supplementary Fig. 10a). In this context, we previously demonstrated reductions in DNA methylation of *Ehhadh* and *Acox1*, two PPARα target genes responsible for fatty-acid β-oxidation in the liver of Wy-offspring[13]. However, they were not listed as DNA hypomethylated genes in Wy-offspring relative to Veh-offspring both at D16 and 14W by MIAMI analysis. Indeed, reductions in DNA methylation of *Ehhadh* and *Acox1* were advanced from D16 toward 14W both in Veh- and Wy-offspring, with no significant difference in DNA methylation status at 14W (Supplementary Fig. 10b, c). These observations suggest that *Ehhadh* and *Acox1* show no epigenetic memory via a DNA methylation mechanism. Then, what determines the epigenetic memory? DNA methylation ratios of *Fgf21* and *Ucp3* at D16 were clearly higher than that of *Ehhadh* and *Acox1*. Therefore, we speculated that high DNA methylation ratios in the suckling period may be related to the epigenetic memory.

In addition to *Fgf21*, many of the 11 PPARα target genes identified as epigenetic memory genes are related to lipid and glucose metabolism such as *Act1*, *Cpt1b*, *Dgat1*, *Fabp3*, *Peci*, *Plin1*, *Psat1*, and *Ucp3*. Because these genes undergo ligand-activated PPARα-dependent DNA demethylation during the suckling period that persists into adulthood, it could be possible that some of the epigenetic memory genes explain the difference in metabolic phenotypes between Wy-HFD and Veh-HFD. However, since the metabolic phenotypes observed in Wy-HFD were alleviated in Wy-FGF21-KO, we speculated that *Fgf21* through DNA demethylation induced during the suckling period may be at least in part associated with the attenuation of diet-induced obesity in adulthood.

Given its unique epigenetic properties, analysis of *Fgf21* methylation status in adulthood may reflect to what extent DNA demethylation is induced in response to milk lipids during the suckling period. In this study, we found that maternal administration of a PPARα ligand during the lactation period promotes DNA demethylation of some PPARα target genes, suggesting that the maternal nutritional environment can be transferred to and stored onto the offspring's genome as epigenetic memory via milk lipids. The data of this study provide the proof of concept of epigenetic modulation in early life through which the epigenetic status of a particular gene can be modified during the suckling period, which provides a critical time window to establish epigenetic memory in the DOHaD hypothesis. It is thus conceivable that epigenetic modulation of metabolic genes by infant formulae and/or functional food, for instance, offers a novel therapeutic strategy to delay the onset of and/or prevent the development of metabolic diseases, such as obesity, in adulthood.

In conclusion, this study provides the first in vivo evidence that FGF21 expression is epigenetically regulated via DNA methylation and that the DNA methylation status, once established in early life, persists into adulthood, thereby contributing to the attenuation of diet-induced obesity (Fig. 9). Thus, we propose that FGF21 could be a key mediator of the developmental programming of obesity.

## Methods

**Animals**. All animal experiments were conducted in accordance with the guidelines of the Tokyo Medical and Dental University Committee on Animal Research, which approved the research protocol (#0170125 A). In each experiment, we purchased pregnant female C57BL6 mice, which were primiparous, from CLEA Japan (Tokyo, Japan) at gestational day 13. After delivery, litter size was adjusted to 5–6 pups (all male) per dam to avoid metabolic drifts due to nutrient availability during lactation. Maternal administration of Wy dissolved in 60% DMSO as Veh was performed at 40 mg/kg body weight (BW)/day during gestation days 14–18 and days 2–16 of the lactation period via intraperitoneal (i.p.) injection[13].

All animals were allowed free access to water and an NCD (#CE-2; 343 kcal/100 g, 12.6% energy as fat; CLEA Japan). In the HFD feeding experiments, 4-week-old mice were fed an HFD (#D12492; 524 kcal/100 g, 60% energy as fat; Research Diets, Inc., New Brunswick, NJ, USA) for 10 weeks (4–14W). Body weight and food intake were measured weekly. We repeated the experiments with the same protocol six times with reproducible results and showed the representative data from one of the six experiments.

PPARα-KO mice[44,45] (strain: B6.129S4-*Pparatm1Gonz*/J, stock number: 008154) were purchased from the Jackson Laboratory (Bar Harbor, ME, USA). Homozygous male and female PPARα-KO mice were crossed to obtain PPARα-KO offspring. Age-matched C57BL6 mice were used as WT control mice.

FGF21-deficient (KO) (strain: C57BL/6NJcl, general. FGF21-KO) mice were kindly provided by Prof. Morichika Konishi in Kobe Pharmaceutical University, Hyogo, Japan[31]. In brief, virgin homozygous male and female (23-week-old) FGF21-KO mice were crossed to obtain FGF21-KO offspring. Maternal administration of Wy dissolved in 60% DMSO as vehicle (Veh) was performed at 40 mg/kg body weight (BW)/day during gestation days 14–18 and 2–16 of the lactation period via intraperitoneal (i.p.) injection. After delivery, litter size was adjusted to five pups (all male) per dam to avoid metabolic drifts due to nutrient availability during lactation. In the HFD feeding experiments, 4-week-old mice were fed HFD for 10 weeks (4–14W). BW and food intake were measured weekly.

**Mass spectrometry for Wy**. For the Wy standard solution, we prepared 0.1 µg/mL Wy dissolved in 50% acetonitrile. As milk samples, we used the gastric contents of pups (Veh- and Wy-offspring), which mainly consisted of milk derived from dams, at D16. We applied Wy standard and milk samples to LC/MS-MS operated in a multiple reaction monitoring mode. According to PubChem, an open chemistry database (https://pubchem.ncbi.nlm.nih.gov/compound/5694#section=Top), the monoisotopic mass (M) values of Wy (pirinixic acid), precursor *m/z* ([M + H]+), and product *m/z* ([M + H]+) are 323.06, 324.06, and 306.04, respectively.

**Lipid composition analysis of mouse milk**. The lipid composition of mouse milk was measured using a capillary GC method[46,47]. In brief, the gastric contents of offspring at D16 were collected and total lipids were extracted using the Bligh and Dyer method with chloroform–methanol (1:1) solution[48]. Fatty acids in total lipids were methylated using boron trifluoride-methanol solution (14% w/v, #B1252, Sigma-Aldrich, Saint Louis, MO, USA), and C8–C24 fatty acids were identified using a capillary gas chromatograph–flame ionization detector (#6890 N, Agilent, Santa Clara, CA, USA) equipped with Omegawax-250 (30 × 0.25 mm, $d_f$ 0.25 µm, #24136, Sigma-Aldrich). C23:0 methyl ester (#91478, Sigma-Aldrich) was used as the internal standard. The coefficient of variation using this method was 0.4–9.9%.

**Fat-free diet administration**. We administered fat-free diet (#D04112303; 0% energy as fat, Research Diets, Inc., New Brunswick, NJ, USA) and control diets (#D12450B; 10% energy as fat, Research Diets, Inc., New Brunswick, NJ, USA) to dams during the late gestation and lactation periods (e14–e18 and D2–D16, respectively, for offspring). At D16, we examined the milk lipid composition in the offspring gastric contents via gas chromatography.

**DNA methylation profiling**. Mouse liver genomic DNA was extracted via the standard proteinase K method. In brief, 50 mg of mouse liver were incubated with 0.5 mL of DNA digestion buffer (50 mM Tris-HCl pH 8.0, 20 mM EDTA pH 8.0, 100 mM NaCl, 1% SDS, 0.8 mg/mL Proteinase K) overnight at 55 °C. Mouse genomic DNA was extracted with neutralized phenol/chloroform/isoamylalcohol (25:24:1) and precipitated with one volume of 100% isopropanol. The DNA precipitate was dissolved in TE buffer and incubated at 65 °C for 15 min to resuspend DNA. Genomic DNA was adjusted to a concentration of 0.1–0.3 μg/μL for later analysis. Human fetal and adult liver genomic DNAs were purchased from Bio-Chain Institute Inc. (Newark, CA, USA) and details about the materials are described in Supplementary Table 3.

MIAMI analysis, a genome-wide analysis of DNA methylation using a gene promoter array and methylation-sensitive restriction enzyme, was performed using a protocol provided by Dr. Izuho Hatada (Gunma University) (http://epigenome.dept.showa.gunma-u.ac.jp/~hatada/miami/image/MIAMI%20Protocol%20V4.pdf)[49]. Briefly, genomic DNAs from two samples were digested with methylation-sensitive *Hpa*II and methylation-insensitive *Msp*I, followed by adaptor ligation and PCR amplification. Amplified DNA from one sample was labeled with Cy3 and the other DNA sample was labeled with Cy5. After labeling, the DNAs were cohybridized to the gene promoter arrays containing 41,332 probes. The *Hpa*II/*Msp*I signal difference was determined as the methylation difference value. Values of <0.65 and >1.4 denoted DNA hypomethylation and hypermethylation, respectively[13,50]. DAVID v6.7 (http://david.abcc.ncifcrf.gov/) was employed for the pathway analysis. The corrected *P*-values were used to judge the candidate genes ($P < 0.05$ was considered significant).

**Bisulfite-sequencing analysis**. Bisulfite-sequencing analysis was performed as follows. To prepare the bisulfite stock solution, 3.8 g of sodium metabisulfite (#31609-45, Nacalai Tesque, Kyoto, Japan), 1.34 g of ammonium sulfite monohydrate (#014-03505, Wako Pure Chemical Industries, Osaka, Japan), and 10 mL of 50% ammonium hydrogen sulfite solution (#014-02905, Wako) were mixed and heated at 80 °C for dissolving. To prepare a working bisulfite solution (8.4 M, pH 5.2–5.3), 12 mL of the bisulfite stock solution were diluted with 2.28 mL of distilled water (dH$_2$O). Two micrograms of the genomic DNA were denatured in 0.3 N NaOH in a total volume of 24.75 μL for 30 min at 37 °C. The sample was mixed with 275 μL of the working bisulfite solution and incubated for 1 h at 70 °C in the dark. DNA was recovered using genomic DNA purification kit (#NPK-101, Toyobo Life Science Department, Osaka, Japan) and dissolved in 100 μL of dH$_2$O. The DNA sample was mixed with 100 μL of 0.4 N NaOH (freshly prepared) and incubated for 15 min at 37 °C. DNA was recovered by adding 150 μL of 5 M ammonium acetate (pH 7.0), 2μL of Ethacinmate (#312-01791, Nippon Gene, Tokyo, Japan), and 750 μL of ethanol. The DNA-containing precipitate was dissolved in 20 μL of 10 mM Tris-HCl/1 mM EDTA (TE, pH 7.5) and subjected to PCR amplification. Sequential PCR amplification of mouse *Fgf21* and human *FGF21*, mouse *Ucp3*, *Ehhadh*, and *Acox1* gene was performed using specific primers described in Supplementary Table 4. The reaction profiles were 40 cycles of 96 °C for 15 s, 59 °C (*Fgf21*), 56 °C (*FGF21*, *Ucp3*), 58 °C (*Ehhadh*) and 64 °C (*Acox1*) for 30 s, and 72 °C for 60 s. The amplified fragments were ligated into pGEM-T easy vectors (#A1360, Promega, Madison, WI, USA), and more than 13 clones were sequenced per reaction. We used a web-based quantification tool for the bisulfite-sequencing analysis of CpG methylation (http://quma.cdb.riken.jp/)[51].

**ChIP assay**. ChIP assays were performed as follows. In brief, 200 mg of frozen livers were homogenized in phosphate-buffered saline (PBS) containing 1% formaldehyde, incubated for 15 min at 37 °C and quenched with 0.125 M glycine. Cross-linked liver tissues were washed in PBS, resuspended in lysis buffer (10% SDS, 50 mM NaCl, 10 mM EDTA, 50 mM Tris-HCl pH 8.0, and protease inhibitors) and sonicated by using a Branson 250 Digital Sonifer (#SFX250, Branson Ultrasonics Corporation, Danbury, CT, USA) at 40% power amplitude. Chromatin samples in 1-mL aliquots were incubated with Protein G-conjugated DynaBeads (#10004D, Life Technologies, Carlsbad, CA, USA) coupled with 5 μg anti-PPARα (a kind gift from Dr. Toshiya Tanaka, Division of Metabolic Medicine, Research Center for Advanced Science and Technology, The University of Tokyo, Japan)[52] or with appropriate antibodies (Supplementary Table 5). The dilution rate for all antibodies is described in Supplementary Table 5. The ChIP-enriched DNA samples were analyzed by quantitative PCR using the primer sets described in Supplementary Table 6.

**Real-time PCR analysis**. Total RNA was isolated from the liver and eWAT at D2, D16, 4W, and 14W. Real-time PCR was performed using the primer sets described in Supplementary Table 7. The mRNA levels were normalized to those of *36B4*, and analyzed using the comparative CT method.

**Biochemical assays**. Serum FGF21 concentrations were determined using Rat/Mouse (#MF2100) and Human (#DF2100) FGF21 enzyme-linked immunosorbent assay (ELISA) kits (Quantikine ELISA; R&D Systems, Minneapolis, MN, USA). According to the manufacturer's instruction, recombinant mouse FGF21 had approximately 21% cross-reactivity in the Human ELISA kit, whereas recombinant human FGF21 had 1.4% cross-reactivity in the Rat/Mouse FGF21 ELISA kit. Serum adiponectin concentrations were determined using a Rat/Mouse Adiponectin

ELISA kit (#MRP300, Quantikine ELISA; R&D Systems). Serum NEFA, TG, and T-Chol levels were measured using NEFA C-Test Wako (#279-75401), TG E-Test Wako (#432-40210), and T-Chol E-Test Wako (#439-17501) kits (Wako Pure Chemical Industries, Ltd., Osaka, Japan), respectively.

**Wy injection**. Adult Wy- and Veh-offspring received a single i.p. injection of Wy in 60% DMSO (1.2 mg/kg BW) and solvent only, respectively. Liver or blood samples were collected 1 and 3 h after the injection to analyze *Fgf21* mRNA expression and serum FGF21 concentrations.

**Glucose tolerance test**. The GTT was performed via an i.p. injection of glucose at 1.0 g/kg body weight, and blood glucose levels were measured before and 30, 60, 90, and 120 min after the injection. Blood glucose was measured using a glucometer (Glutest PRO R; Sanwa Kagaku Kenkyusho Co., Ltd., Aichi, Japan). Serum insulin levels were measured using an ELISA kit (Ultra Sensitive Mouse Insulin ELISA Kit, Morinaga Institute of Biological Science, Inc., Kanagawa, Japan).

**Recombinant human (rh) FGF21 administration**. Four-week-old male WT mice were housed at 25 °C and fed an HFD. rhFGF21 was purchased from PeproTech (#100-42; Rocky Hill, NJ, USA). The bioactivity of rhFGF21 was roughly comparable to that of recombinant mouse FGF21 in mice[53,54]. Mice were administered either saline or 0.4 mg/kg BW recombinant human FGF21 ($n = 8$/group) once daily for 4 weeks.

**Histological analysis**. The epididymal white adipose tissue (eWAT) was fixed with neutral-buffered formalin and embedded in paraffin. Then, 5-μm thick sections were stained with hematoxylin and eosin (HE). To measure adipocyte cell size, more than 200 cells were counted per section using image-analyzing software (WinRoof, Mitani, Tokyo, Japan). The quantitative histological analysis was performed by three investigators, who had no knowledge of the slide origin.

**Statistical analysis**. Data are expressed as the mean ± standard error of the mean (SEM). Data were compared using a chi-squared test and Student's *t*-test. Comparison of body weight difference was evaluated by two-way analysis of variance (ANOVA) (between-group, within-time, and interaction of time and group). Furthermore, comparison between groups at each week of age was evaluated by multiple *t*-test with Bonferroni correction. Spearman's rank correlation coefficient was used to evaluate correlations between variables. $P < 0.05$ was considered statistically significant. Statistical analysis was performed using Prism 6 (GraphPad Software, Inc., La Jolla, CA, USA).

**Data availability**. The data that support the findings of this study are available from the corresponding author upon reasonable request.

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

## Acknowledgements

We thank Ms. Rie Yamada and Ms. Yumi Gotoda for secretarial assistance and Drs. Mayumi Takahashi (Osaka Women's Junior College) and Yoshiaki Nakayama (Kobe Pharmaceutical University) for technical assistance. We also thank Dr. Toshiya Tanaka (Division of Metabolic Medicine, Research Center for Advanced Science and Technology, The University of Tokyo, Japan) for providing anti-PPARα antibody. This work was supported in part by Grants-in-Aid for Scientific Research (KAKENHI) from the Japan Society for the Promotion of Science [grant numbers JP16H05331 (Y.O.), JP16H01354 (Y.O.), JP26461376 (K.H.), and JP26350884 (X.Y.)]; Secom Science and Technology Foundation (Y.O.); The Naito Foundation (Y.O.), Nestlé Nutrition Council (X.Y.), Japan; Takeda Science Foundation (K.H.); Dairy Products Health Science Council of Japan Milk Academic Alliance (K.H.); and Tohoku Medical Megabank Project (T.T.-I.).

## Author contributions

X.Y., K.T. and K.H. conceived the project, designed, and performed the experiments, and evaluated the data; Y.O. conceived the project and designed the experiments; K.K., N.H., T.S. and M.H. participated in and contributed to all in vitro and in vivo experiments; T.E. and I.H. contributed to the MIAMI analysis; M.N. performed LC/MS-MS analysis; Y.K. performed milk lipid analysis; M.K. and N.I. contributed to the analysis of FGF21-KO mice; Y.N. and H.S. contributed to the analysis of PPARα-KO mice; T.T.-I. contributed to the statistical analysis; T.E. and Y.K. evaluated the data and participated in preparing the manuscript; K.H. and Y.O. wrote the paper and supervised the entire project.
