## [Peer Review File · Nature Communications]

Reviewers' comments:

Reviewer #1 (Expert in obesity and FGF21 biology; Remarks to the Author):

This study provides evidence demonstrating that activation of PPARalpha by its agonist causes the demethylation of the FGF21 gene, and such an epigenetic modification can be transferred to the next generation, thereby causing increased FGF21 expression and partial resistance to diet-induced obesity in the offspring. The finding that epigenetic modification of FGF21 in the perinatal mice is related to reduced dietary obesity in adulthood is novel and potentially important. Nevertheless, the conclusions can be further consolidated by additional experiments.

Major points:

1. Although the data shows clearly that PPARalpha causes demethylation of the FGF21 gene, the underlying mechanism remains unclear. How does PPARalpha, as a transcription factor, regulate the methylation status of the gene promoters?
2. It is interesting that, although PPARalpha-dependent demethylation also occurs in other target genes (such as those genes involved in fatty acid oxidation), not all of them causes epigenetic memory. Please explain why only a portion of the PPARalpha target genes are classified as epigenetic memory genes.
3. Fig. 4e: the difference was observed only in a single time point (24 hour). To avoid concerns of coincident findings, serum FGF21 in 18 hours after fasting should also be measured, if samples are still available.
4. Fig. 5h: the circadian rhythm changes of FGF21 are unclear to me. Why does FGF21 keep declining during the observation period?
5. The body weight changes shown in Fig. 5B is really marginal between the two groups. Why the difference occurs only in eWAT, but not other adipose depots and liver (Fig. 5B)?
6. The FGF21 treatment study in Figure 6 does not really help to strengthen the conclusion that offspring mice with high FGF21 inherited from Wy-treated parental mice protects against obesity. Ideally, an in vivo knocking-down approach (FGF21 neutralization) should be used to investigate whether the obesity-resistant phenotype of Wy-treated offspring can be reversed by blockage of FGF21 increase.

Minor point:

Please state the source of human fetal and adult livers used in this study (supplemental figure 3a).

Reviewer #2 (Expert in PPARs and metabolism; Remarks to the Author):

In the present paper, Yuan and colleagues describe how administering a PPARalpha ligand can induce epigenetic modulation of fibroblast growth factor 21 (fgf21) in the liver of perinatal mice during the period of lactation. They show that DNA demethylation of the fgf21 gene persists into adulthood and link this epigenetic memory to differences metabolic phenotypes following high-fat feeding. They conclude that milk lipids, as PPARalpha ligands, could induce DNA demethylation of fgf21 and thereby contribute to the attenuation of diet-induced obesity.

In an article published in *Diabetes* in 2015 (Ligand-Activated PPAR α -Dependent DNA Demethylation Regulates the Fatty Acid β -Oxidation Genes in the Postnatal Liver) the authors of this paper perform an analysis of methylation by isoschizomers on liver tissue derived from offspring originating from dams that received the synthetic PPAR α ligand Wy14643, in order to identify genes that become hypomethylated in response to this ligand. In the present paper, the authors repeat the same strategy and identify 11 other genes that are known PPAR α target genes, of which they choose to focus on *fgf21*.

Major comments:

1) In the first part of the paper, the authors try to determine whether *fgf21* can be regulated through epigenetic modifications and which period from gestation to adulthood is the critical time-window for the induction of epigenetic memory. In the second part of the paper, they describe differences in metabolic phenotypes in offspring from control dams (receiving vehicle) versus dams that received Wy14643. Eventually, the authors try to link the two parts together by concluding that epigenetic regulation of *fgf21* solely is responsible for attenuating the phenotype of diet-induced obesity, disregarding the possible effect of the other genes that were identified to be regulated through PPAR α -mediated epigenetic memory. It seems far-fetched to attribute the observed differences in metabolic phenotype solely to epigenetic modulation of *fgf21*. The fact that *fgf21* administration itself attenuates the phenotype of diet-induced obesity was already firmly established by other studies and does not allow for any inferences about the direct effect of epigenetic modulation of *fgf21* on diet-induced obesity. Hence, the title of the paper oversells the actual data. Instead, it should be stated that epigenetic modulation of *fgf21* in the perinatal mouse liver is associated with attenuation of diet-induced obesity in adulthood.

2) The authors propose that the critical time window of epigenetic *fgf21* modulation is during lactation, as they see no induction of *fgf21* demethylation in adult mice that receive Wy14643 administration, nor in the offspring of pregnant dams that received Wy14643 administration during the late gestation period. However, they do not provide any data or explanation as to why the critical window of epigenetic *fgf21* modulation is limited to this period. The fact that *fgf21* demethylation seems to be limited to lactation, and is not induced during gestation, indeed suggests that the main inducing factor of epigenetic modulation is present in the mother milk. However, it was not measured whether Wy14643 itself ends up in the mother milk, whether it is able to change the lipid composition and concentrations, or whether it might induce its effects through a synergistic reaction between Wy14643 and natural milk PPAR α ligands. Furthermore, if it is the case that solely the presence of Wy14643 in mother milk is able to induce PPAR α -mediated epigenetic memory, it cannot be concluded that the PPAR α ligands that are naturally present in mother milk could induce the same effect to a similar level. Therefore, to be able to draw the conclusions that were drawn in the present paper, it is needed to show that natural milk lipids, in physiological concentrations, are able to induce epigenetic modulation of *fgf21*. In addition, measuring the lipid composition and Wy14643 levels in the mother milk, or including an extra control group where Wy14643 is administered directly to pups during lactation instead of via the mother milk, could give more insight into whether the epigenetic modulation depends on milk-derived factors, or solely on Wy14643 administration. Furthermore, it could indicate whether the lactation period is indeed the critical window of epigenetic modulation, or whether this is only suggested because epigenetic modulation depends on mother milk-specific synergistic mechanisms.

3) There is no mechanistic framework on how PPAR α activation may influence the methylation status of genes. In their previous paper, the authors speculate "that the ligand-activated PPAR α recruits factors involved in DNA demethylation to the promoter regions of the fatty acid β -oxidation genes via the PPAR-responsive element, thus leading to the gene-specific DNA demethylation." One would expect inclusion of some experimental evidence on the actual mechanism of PPAR α -mediated demethylation in the present paper.

Minor comments:

1) In figure 2a, the BS region of *fgf21* is shown. As stated by the authors, two PPAR response elements (PPRE) were found near the TSS of the gene. However, the BS region of *fgf21* only includes the second PPRE (100 bp upstream of the TSS) that was found *in silico*, and no methylation status is measured for the first PPRE (1000 bp upstream of the TSS), meaning that the role of the first PPRE is not taken into account in the bisulfite analysis and further analyses. Please speculate on the potential consequences of excluding the first PPRE.

2) In the text concerning figure 3, it could be stated more clearly that periods i and ii actually consist of two distinct experiments. Furthermore, no explanation is given as to why Wy14643 might not be able to induce epigenetic modulations of *fgf21* during adulthood.

3) In figure 4, large differences in mRNA expression of *fgf21* and serum FGF21 levels can be seen between D2 and D16, although both periods fall in the period of lactation that was defined as the critical window of epigenetic *fgf21* modulation by the authors. Please provide a potential explanation for the decreased expression and serum levels at D16 compared to D2.

4) The legend of figure 4d mentions 3-8 animals per group. However, the right panel (correlation) of figure 4d shows only 3-4 mice per group.

5) Performing a correlation analysis for $n=7-8$ is meaningless

6) Figure 4c and 4d: Can the authors explain how it is possible that *fgf21* plasma levels are much more highly induced upon Wy injection as opposed to *fgf21* mRNA in liver. It suggests that a significant portion of plasma *fgf21* is not derived from liver.

7) At figure 5b, significant differences that are indicated for the body weight between Veh-HFD and Wy-HFD groups in week 11 to 14 seem very small (the graphs indicate differences around 2-3 grams). From our experience, it seems doubtful that these differences can be significant in an experiment conducted with 11 mice per group. Please verify statistical significance.

General recommendation: The conclusions that are drawn in the present paper seem heavily based on correlations and speculations. The authors should indicate whether *fgf21* epigenetic memory could be induced specifically during the lactation period in natural conditions or whether only the administration of Wy14643 is able to induce epigenetic modulation, and provide more substantial evidence that the induction of *fgf21* demethylation is indeed largely responsible for the attenuation of diet-induced obesity, or otherwise tone down the conclusions that are drawn.

Reviewer #3 (Experts in epigenetics and obesity; Remarks to the Author):

NCOMMS-17-05439

The authors report an examination of DNA methylation and expression dynamics of FGF21 in peri and postnatal mouse liver and correlate these findings to attenuation of diet-induced obesity in adulthood. The study is interesting and is overall well written. The data measures and analysis appear well performed and are appropriate. It suggests DNA-methylation dynamics associated with Ppara binding near the FGF21 promoter early in life can program FGF21 expression stably and over the long term. The data are of interest but still fall short of convincing the reader on all aspects particularly regarding technical methodology and mechanism. If the authors can address key points below, the ideas and experiments presented here would be appropriate for this journal.

My Concerns:

Specifically, it is not entirely clear that this is FGF21 mediated. If FGF21 mediated, the effects of the intervention should be abolished in either liver FGF21 knockout animals, or through chronic immunoneutralization of FGF21 in control and reprogrammed animals. Can these experiments be done?

Second, the epigenetic mechanism is not entirely convincing. In the end, the authors are proposing that an ~15 % change in DNA-methylation is responsible for the changes in FGF21 expression. How do the authors proposed this results in altered gene regulation. Which CpG sites are involved? How do they regulate FGF21 expression? How is transcription and transcription factor binding so reproducibly affected, by these modest changes? The DNA methylation correlation to FGF expression on the mouse to mouse basis is nice. Are there specific CpG's where DNA methylation changes correlate poorly to the severity of or FGF expression? Are there others where this correlation is perfect? Is there cellular heterogeneity in expression of FGF21 in the liver, does this change with the treatments (ie. the bisulfite shows that some cells have a fully methylated sequence and others fully unmethylated).

The phenotype is very mild, which is fine. That said, the chance that this is simple variation between individuals is thus enhanced. The third important concern is therefore that there is little description of the exact mouse husbandry and animal choice that was performed. Metabolic changes can be observed between mice born of different littersizes, or born to different fathers, or born to virgin vs multiparous mothers. How many of these variables were controlled and how. In the end, the key features of the study must be repeated with maternally matched animals ie. either virgin or multiparous mothers, comparing only equally sized litters. Were all the litter sizes equal? Overall the methods section suggests that the experiment was done in one large cohort... If so this needs to be repeated.

It is not clear to me the mechanism by which Wy is exerting its effect, direct (ie. passing through the milk) or indirectly (affecting the mothers metabolism and thus the composition of the milk). This should be addressed. Can the authors measure Wy in the milk? Has it been ruled out that the Wy compound is working directly in the offspring ie. getting into the milk? Alternatively, can the authors identify a milk component that can mimic the effect?

Additional concerns:

Figure 1 – please show all the data. A volcano plot should be shown for each D6 and 14W timepoints to give an indication of the spread and noise of the data. A correlation plot showing the differences at 14W (y-axis) vs. D6 (x-axis) to show the correlation between the datasets should be shown. Significantly hyper and hypo methylated alleles should be highlighted on the plot. A heatmap would give important information of the noise and variation of the data for instance.

Throughout the manuscript the authors talk about DNA demethylation. While not entirely incorrect, the most common use of the term in the literature implies an active process. Since the authors provide no evidence of active DNA demethylation, they should refer throughout to reductions or losses in DNA methylation.

The quantification of adipocyte size appear not to be normalized properly. The area under the curve should be equal adding up to 100% in both curves. This doesn't appear to have been done in Figure 5 and Supp Fig 5. This analysis should be redone and the median value and interquartile range highlighted. Statistical analysis based a number of mice should also be applied.

Figure 3. I realize it is substantial work but I'm not sure I agree that the DNA methylation is

unchangeable in the adult. The Wy experiment in 3b should be repeated with a longer intervention. There is a clear (non-significant change in the figure shown). Do these authors really believe this couldn't be altered in adulthood by Wy?

Similarly, the authors should perform more replicates of the data in Supp Fig b-d. There are trends in H3K27me3 and Ppara ChIP that could be very important if they are real. These data should be done with many replicates and included in the main Figure. I can very much imagine decreased K27me3 and enhanced Ppara binding at the locus even constitutively. It is important not to hide this in the supplements or state no change, just because the error is substantial. Overall the authors appear to have done very precise work with very tight error where they want to see a difference. They should strive for the same here.

Minor concerns:

One additional suggestion, that perhaps is for the future for the authors. Mice compensate for high-fat diet extremely well if HFD is started at a young age. Here the authors have started HFD at the youngest possible age when the mice are only half grown. The authors may find more dramatic metabolic shifts if HFD is first applied after 10 weeks of age.

The methods section could use more details. There is a 'mm' typo in the histology section. How many sections and how far apart were the sections counted from each mouse?

Given the time-frame of best induction, I'm not sure peri-natal is the best term. Perhaps post-natal?

To Reviewer #1

Thank you very much for your valuable comments and suggestions. According to your comments and suggestions, we have revised the manuscript as follows.

General comments

This study provides evidence demonstrating that activation of PPAR α by its agonist causes the demethylation of the FGF21 gene, and such an epigenetic modification can be transferred to the next generation, thereby causing increased FGF21 expression and partial resistance to diet-induced obesity in the offspring. The finding that epigenetic modification of FGF21 in the perinatal mice is related to reduced dietary obesity in adulthood is novel and potentially important. Nevertheless, the conclusions can be further consolidated by additional experiments.

Major point 1

Although the data shows clearly that PPAR α causes demethylation of the FGF21 gene, the underlying mechanism remains unclear. How does PPAR α , as a transcription factor, regulate the methylation status of the gene promoters?

Our response

A couple of studies proposed that transcriptional factors such as PPAR γ and aryl hydrocarbon receptor (Ahr) induce DNA demethylation by Ten-eleven translocation (TET) enzymes (TET1, 2 and 3) (*Nat Commun* 4, e2262, 2013, *Sci Rep* 6, e34989, 2016). In this study, we hypothesized that TET enzymes are related to DNA demethylation of *Fgf21* via PPAR α . By ChIP assays for TET1-3, we found that TET2 but rather than TET 1 and 3 was recruited abundantly to the promoter region of *Fgf21* in Wy-offspring relative to Veh-offspring at D16, suggesting that TET2 is involved in DNA demethylation of *Fgf21*. On the other hand, there was no significant difference in the recruitment of DNA methyltransferases (DNMTs: DNMT3a and 3b) between Wy- and Veh-offspring at D16, suggesting that DNMTs are not related to PPAR α -dependent DNA demethylation.

In the revised manuscript, we have included the data on ChIP analysis as **Figure 5** (newly assigned) and added the following sentences in **Results** section (at pages 14-15 on lines 278-296),

“To provide an insight into the molecular mechanism of PPAR α -dependent DNA demethylation of *Fgf21* during lactation period, we evaluated mRNA expressions for epigenetic modifiers such as TET enzymes and DNA methyltransferases (DNMTs). Postnatal ontogenic gene expression of TET enzymes showed that *Tet1* mRNA levels were decreased in a time-dependent manner after birth, whereas both *Tet2* and *Tet3* mRNA levels were increased with a peak at D16 and declined thereafter. On the other hand, both *Dnmt3a* and *Dnmt3b* mRNA levels were gradually decreased toward adulthood (Fig. 5a). We performed ChIP assays for these epigenetic modifiers at D16; TET2 but not TET1 or TET3 was recruited abundantly to the promoter region of *Fgf21* in Wy-offspring than in Veh-offspring (Fig. 5b).

On the other hand, there was no significant difference in the recruitment of DNMT3a and DNMT3b to the promoter region of *Fgf21* between Wy- and Veh-offspring at D16 (Fig. 5b). In this study, adult WT mice, when treated directly with Veh (Veh-mice) or Wy (Wy-mice) for 2 weeks after the suckling period (4–6W), showed no significant difference in DNA methylation status of *Fgf21* between 4W and 6W (Fig. 3b). At 6W, there was no significant difference in the recruitment of TET2 to the promoter region of *Fgf21* between Veh- and Wy-mice (Fig. 5c).”

We also added the statement on the mechanical insight into PPAR α -dependent DNA demethylation of *Fgf21* in **Discussion** section (at pages 22-23 on lines 446-464),

“The detailed mechanism underlying PPAR α -dependent DNA demethylation of its target genes remains to be elucidated. A couple of studies proposed that transcriptional factors such as PPAR γ and aryl hydrocarbon receptor (Ahr) induce DNA demethylation by TET enzymes. In this study, we hypothesized that TET enzymes are related to DNA demethylation of *Fgf 21* via PPAR α . Given that TET2 but not TET1 or TET3 were significantly more abundantly recruited to the promoter region of *Fgf21* in Wy-offspring than in Veh-offspring at D16, TET2, an eraser of DNA methylation, may be involved in DNA demethylation of *Fgf21* during the suckling period. Furthermore, under PPAR α deficiency, the recruitment of TET2 to the promoter region of *Fgf21* was roughly equivalent between Wy- and Veh-offspring (Supplementary Fig. S2b), thereby

indicating the potential interaction between TET2 and PPAR α . In adulthood, we found no differences in the recruitment of TET2 between the mice treated with vehicle (Veh-mice) and those treated with Wy (Wy-mice) (Fig. 5c), which may explain that PPAR α -dependent DNA demethylation of *Fgf21* occurs only during the suckling period, a critical time window for DNA demethylation. On the other hand, DNMT3a and 3b, which are *de novo* DNMTs; writers of DNA methylation may not be related to PPAR α -dependent DNA demethylation.”

Major point 2

It is interesting that, although PPARalpha-dependent demethylation also occurs in other target genes (such as those genes involved in fatty acid oxidation), not all of them causes epigenetic memory. Please explain why only a portion of the PPARalpha target genes are classified as epigenetic memory genes.

Our response

In MIAMI analysis, we identified 11 PPAR α target genes, that were DNA hypomethylated in Wy-offspring relative to Veh-offspring both at D16 and 14W (**Table 1**). We investigated transcriptional factor binding motifs in the promoter regions of the 11 genes and found no common consensus motifs besides the PPAR response element (PPRE) (data not shown). Since the microarray in MIAMI analysis only contains 60-mer-portions of *HpaII* fragments located in promoter regions, it does not necessarily cover all the CpG sites in genes. Therefore, the results of MIAMI analysis may reflect DNA methylation status in a limited region of genes, which may sometimes make a difference with the results of bisulfite sequencing to target a long region containing the CpG sites.

In this study, we performed bisulfite sequencing for *Ucp3* and found that DNA methylation status of *Ucp3* at D16 does not change at 14W and that reductions in DNA methylation of *Ucp3* are enhanced in Wy-offspring relative to Veh-offspring at D16, showing that the difference in DNA methylation between Wy- and Veh-offspring was maintained at D16 and 14W, which was similar to that with *Fgf21* (**Supplementary Figure S10a** in the revised manuscript). In this context, we previously demonstrated reductions in DNA methylation of *Ehhadh* and *Acox1*, two PPAR α target genes responsible for fatty acid β -oxidation in the liver of Wy-offspring (*Diabetes* **64**, 775-784, 2015). However, they were not listed as DNA hypomethylated genes in

Wy-offspring relative to Veh-offspring both at D16 and 14W by MIAMI analysis. Indeed, reductions in DNA methylation of *Ehhadh* and *Acox1* were advanced from D16 toward 14W both in Veh- and Wy-offspring, with no significant difference in DNA methylation status at 14W (**Supplementary Figure S10b, c**). These observations suggest that *Ehhadh* and *Acox1* show no epigenetic memory via a DNA methylation mechanism. Then, what determines the epigenetic memory? DNA methylation ratios of *Fgf21* and *Ucp3* at D16 were clearly higher than that of *Ehhadh* and *Acox1*. Therefore, we speculated that high DNA methylation ratios in the suckling period may be related to the epigenetic memory.

We added the statement above in **Discussion** section in the revised manuscript (at pages 28-29 on lines 560-588).

Major point 3

Fig. 4e: the difference was observed only in a single time point (24 hour). To avoid concerns of coincident findings, serum FGF21 in 18 hours after fasting should also be measured, if samples are still available.

Our response

We repeated the experiment and measured serum FGF21 and NEFA concentrations (**Figure 4e and 4f** in the original manuscript) in 18 hours after fasting. We added the data as **Figure 6e and 6f** in the revised manuscript.

Major point 4

Fig. 5h: the circadian rhythm changes of FGF21 are unclear to me. Why does FGF21 keep declining during the observation period?

Our response

FGF21 exhibits a circadian oscillation with a peak of expression correlating with that of free fatty acids (*Clin Chem* **57**, 691–700, 2011), as we also observed in **Supplementary Figure S7g**. On the other hand, a recent study demonstrated that circadian oscillation of FGF21 is disrupted with increased dietary fat (*J Nutr Biochem* **40**, 116–121, 2017), which is consistent with our current observation (**Figure 7h; Figure 5h** in the original manuscript). In this study, under high-fat diet feeding, FGF21 mRNA and protein levels keep declining from ZT 0 to ZT 12 and are thereafter gradually increased with a peak during the dark phase in the liver. The disrupted oscillation under high dietary fat is

congruent with that of some adipocytokines such as adiponectin (*Endocrinology* **150**, 161–168, 2009; *Obesity* (Silver Spring) **8**, 230–238, 2010; *Cell Metab* **6**, 414–421, 2007). However, it remains unclear why high-fat diet disrupts circadian oscillation of FGF21.

Major point 5

The body weight changes shown in Fig. 5B is really marginal between the two groups. Why the difference occurs only in eWAT, but not other adipose depots and liver (Fig. 5B)?

Our response

Following the reviewer's comment, we evaluated mRNA expression of *Egr1* and *c-fos*, which are known to be downstream of *Fgf21* in iWAT and BAT (**Supplementary Figure S8c, d**). We found that mRNA expression of the above genes is roughly equivalent between Wy-HFD and Veh-HFD in both fat depots. In a previous study, when 10-week-old mice are injected with 1 or 10 mg/kg body weight/day recombinant mouse (rmu) FGF21 twice a day for 21 days (*PLoS One* **7**, e40164, 2012), eWAT weight but not iWAT weight was significantly reduced, which is consistent with our current data. In addition, 2 hours after a single injection of 0.6 mg/kg recombinant human FGF21, expression of *Egr1* and *c-fos* mRNA was more augmented in eWAT than in iWAT (*J Biol Chem* **291**, 10867-10875, 2016). Together with our current data, these observations suggest the selective response of eWAT vs. iWAT to FGF21, although its molecular mechanism remains unclear. We have made comment on the selective response of fat depots in response to FGF21 in **Results** section (at page 19 on lines 383-386) as follows;

“The mRNA expression of these genes was roughly equivalent between Wy-HFD and Veh-HFD both in iWAT and BAT (Supplementary Fig. S8c, d), suggesting the selective response of fat depots in response to FGF21.”

Major point 6

The FGF21 treatment study in Figure 6 does not really help to strengthen the conclusion that offspring mice with high FGF21 inherited from Wy-treated parental mice protects against obesity. Ideally, an in vivo knocking-down approach (FGF21 neutralization) should be used to investigate whether the

obesity-resistant phenotype of Wy-treated offspring can be reversed by blockage of FGF21 increase.

Our response

Following your comment, we obtained FGF21-deficient (KO) (strain: C57BL/6NJcl, global knockout) mice, which are kindly provided by Prof. Morichika Konishi in Kobe Pharmaceutical University, Hyogo, Japan (*Endocrinology* **150**, 4625-4633, 2009) and induced DNA demethylation of all the PPAR α target genes except *Fgf21* by maternal administration of Wy. We examined the metabolic phenotype of the offspring derived from FGF21-KO dams treated with Wy and Veh during 10 weeks of HFD feeding (4–14W). We included the data as **Figure 8** (newly assigned in the revision) and added the following sentences in **Results** section (at pages 20-21 on lines 410-424),

“We obtained FGF21-deficient (KO) mice and induced DNA demethylation of all the PPAR α target genes except *Fgf21* by maternal administration of Wy (Fig. 8a). We examined the metabolic phenotype of the offspring derived from FGF21-KO dams treated with Wy and Veh during 10 weeks of HFD feeding (4–14W) (Veh-FGF21-KO and Wy-FGF21-KO, respectively) (Fig. 8a). As shown in Figure 8b, we found no significant difference in body weight between Veh-FGF21-KO and Wy-FGF21-KO. The weights of iWAT, eWAT, and BAT were roughly comparable between the Veh-FGF21-KO and Wy-FGF21-KO at 14W (Fig. 8c). Serum FGF21 was not detected in both groups, proving systemic FGF21 deficiency (Fig. 8d). Histologically, adipocyte cell size in eWAT appeared to be comparable between the Veh-FGF21-KO and Wy-FGF21-KO groups at 14W (Fig. 8e). Consistently, the mRNA expression of *Egr1*, *c-fos*, *Hsl* and *Atgl* showed no significant difference between the two groups (Fig. 8f). These observations are consistent with the notion that *Fgf21* plays a major role in the metabolic phenotypes of Wy-HFD (Fig. 7).”

We also added the following sentences in **Discussion** section (at page 29 on lines 595-599),

“However, since the metabolic phenotypes observed in Wy-HFD were cancelled in Wy-FGF21-KO mice, we speculated that *Fgf21* through DNA demethylation induced during the suckling period may be at least in part associated with the attenuation of diet-induced obesity in adulthood.”

Minor point

Please state the source of human fetal and adult livers used in this study (supplemental figure 3a).

Our response

We uploaded the information on the source of human fetal and adult liver genomic DNAs purchased from BioChain Institute Inc. (Newark, CA, USA) in **Supplementary Table S3**. We have made comment on the source of human fetal and adult liver genomic DNAs in **Methods** section (at page 32 on lines 661-663) as follows;

“Human fetal and adult liver genomic DNAs were purchased from BioChain Institute Inc. (Newark, CA, USA) and details of the materials were described in Supplementary Table S3.”

To Reviewer #2

Thank you very much for your valuable comments and suggestions. According to your comments and suggestions, we have revised the manuscript as follows.

General comments

In the present paper, Yuan and colleagues describe how administering a PPAR α ligand can induce epigenetic modulation of fibroblast growth factor 21 (fgf21) in the liver of perinatal mice during the period of lactation. They show that DNA demethylation of the fgf21 gene persists into adulthood and link this epigenetic memory to differences metabolic phenotypes following high-fat feeding. They conclude that milk lipids, as PPAR α ligands, could induce DNA demethylation of fgf21 and thereby contribute to the attenuation of diet-induced obesity.

In an article published in Diabetes in 2015 (Ligand-Activated PPAR α -Dependent DNA Demethylation Regulates the Fatty Acid β -Oxidation Genes in the Postnatal Liver) the authors of this paper perform an analysis of methylation by isoschizomers on liver tissue derived from offspring originating from dams that received the synthetic PPAR α ligand Wy14643, in order to identify genes that become hypomethylated in response to this ligand. In the present paper, the authors repeat the same strategy and identify 11 other genes that are known PPAR α target genes, of which they choose to focus on fgf21.

Major point 1

1) In the first part of the paper, the authors try to determine whether fgf21 can be regulated through epigenetic modifications and which period from gestation to adulthood is the critical time-window for the induction of epigenetic memory. In the second part of the paper, they describe differences in metabolic phenotypes in offspring from control dams (receiving vehicle) versus dams that received Wy14643. Eventually, the authors try to link the two parts together by concluding that epigenetic regulation of fgf21 solely is responsible for attenuating the phenotype of diet-induced obesity, disregarding the possible effect of the other genes that were identified to be regulated through PPAR α -mediated epigenetic memory. It seems far-fetched to attribute the observed differences in metabolic phenotype solely to epigenetic modulation of fgf21. The fact that fgf21

administration itself attenuates the phenotype of diet-induced obesity was already firmly established by other studies and does not allow for any inferences about the direct effect of epigenetic modulation of fgf21 on diet-induced obesity. Hence, the title of the paper oversells the actual data. Instead, it should be stated that epigenetic modulation of fgf21 in the perinatal mouse liver is associated with attenuation of diet-induced obesity in adulthood.

Our response

Following your comment, we obtained FGF21-deficient (KO) (strain: C57BL/6NJcl, global knockout) mice, which are kindly provided by Prof. Morichika Konishi in Kobe Pharmaceutical University, Hyogo, Japan (*Endocrinology* **150**, 4625-4633, 2009) and induced DNA demethylation of all the PPAR α target genes except *Fgf21* by maternal administration of Wy. We examined the metabolic phenotype of the offspring derived from FGF21-KO dams treated with Wy and Veh during 10 weeks of HFD feeding (4–14W). We included the data as **Figure 8** (newly assigned in the revision) and added the following sentences in **Results** section (at pages 20-21 on lines 410-424),

“We obtained FGF21-deficient (KO) mice and induced DNA demethylation of all the PPAR α target genes except *Fgf21* by maternal administration of Wy (Fig. 8a). We examined the metabolic phenotype of the offspring derived from FGF21-KO dams treated with Wy and Veh during 10 weeks of HFD feeding (4–14W) (Veh-FGF21-KO and Wy-FGF21-KO, respectively) (Fig. 8a). As shown in Figure 8b, we found no significant difference in body weight between Veh-FGF21-KO and Wy-FGF21-KO. The weights of iWAT, eWAT, and BAT were roughly comparable between the Veh-FGF21-KO and Wy-FGF21-KO at 14W (Fig. 8c). Serum FGF21 was not detected in both groups, proving systemic FGF21 deficiency (Fig. 8d). Histologically, adipocyte cell size in eWAT appeared to be comparable between the Veh-FGF21-KO and Wy-FGF21-KO groups at 14W (Fig. 8e). Consistently, the mRNA expression of *Egr1*, *c-fos*, *Hsl* and *Atgl* showed no significant difference between the two groups (Fig. 8f). These observations are consistent with the notion that *Fgf21* plays a major role in the metabolic phenotypes of Wy-HFD (Fig. 7).”

We also added the following sentences in **Discussion** section (at page 29 on lines 595-599),

“However, since the metabolic phenotypes observed in Wy-HFD were cancelled in Wy-FGF21-KO mice, we speculated that *Fgf21* through DNA demethylation induced during the suckling period may be at least in part associated with the attenuation of diet-induced obesity in adulthood.”

Nonetheless, we agree with you that other epigenetic memory genes besides *Fgf21*, such as *Ucp3*, could contribute to the attenuation of diet-induced obesity in adulthood, and have changed the title of the manuscript as ‘**Epigenetic modulation of fibroblast growth factor 21 in the perinatal mouse liver is associated with the attenuation of diet-induced obesity in adulthood.**’

Major point 2

*2)-1. The authors propose that the critical time window of epigenetic *fgf21* modulation is during lactation, as they see no induction of *fgf21* demethylation in adult mice that receive Wy14643 administration, nor in the offspring of pregnant dams that received Wy14643 administration during the late gestation period. However, they do not provide any data or explanation as to why the critical window of epigenetic *fgf21* modulation is limited to this period.*

Our response

We previously reported genome-wide methylation changes in the postnatal mouse liver, comparing e18.5 with D2, D2 with D16, and D16 with D28 (*Diabetes* **64**, 775-784, 2015). By the MIAMI analysis, we found marked changes in DNA methylation from D2 to D16 but marginally from e18.5 to D2 or from D16 to D28. These observations suggest the presence of epigenetic or DNA methylation plasticity in the liver specifically during the suckling period.

In the revised manuscript, we show that Ten-eleven translocation (TET) 2 was recruited abundantly to the promoter region of *Fgf21* in Wy-offspring relative to Veh-offspring at D16, whereas at 6W in adulthood, the recruitment of TET2 to the promoter region of *Fgf21* was roughly equivalent between the mice treated with vehicle (Veh-mice) and those treated with Wy (Wy-mice), which may explain that PPAR α -dependent DNA demethylation of *Fgf21* occurs only during the suckling period. Accordingly, we have included the data on ChIP analysis as **Figure 5** (newly assigned)

with the statement on the critical window for PPAR α -dependent DNA demethylation of *Fgf21* in **Results** section (at page 15 on lines 286-296) as follows,

“We performed ChIP assays for these epigenetic modifiers at D16. At D16, TET2 but not TET1 or TET3 was recruited abundantly to the promoter region of *Fgf21* in Wy-offspring relative to Veh-offspring (Fig. 5b). On the other hand, there was no significant difference in the recruitment of DNMT3a and DNMT3b to the promoter region of *Fgf21* between Wy- and Veh-offspring at D16 (Fig. 5b). In this study, adult WT mice, when treated directly with Veh (Veh-mice) or Wy (Wy-mice) for 2 weeks after the suckling period (4–6W), showed no significant difference in DNA methylation status of *Fgf21* between at 4W and 6W (Fig. 3b). At 6W, there was no significant difference in the recruitment of TET2 to the promoter region of *Fgf21* between Veh- and Wy-mice (Fig. 5c).”

We also added the following sentences in **Discussion** section (at pages 22-23 on lines 446-462),

“The detailed mechanism underlying PPAR α -dependent DNA demethylation of its target genes remains to be elucidated. A couple of studies proposed that transcriptional factors such as PPAR γ and aryl hydrocarbon receptor (Ahr) induce DNA demethylation by TET enzymes. In this study, we hypothesized that TET enzymes are related to DNA demethylation of *Fgf 21* via PPAR α . Given that TET2 but not TET1 or TET3 were significantly more abundantly recruited to the promoter region of *Fgf21* in Wy-offspring than in Veh-offspring at D16, TET2, an eraser of DNA methylation, may be involved in DNA demethylation of *Fgf21* during the suckling period. Furthermore, under PPAR α deficiency, the recruitment of TET2 to the promoter region of *Fgf21* was roughly equivalent between Wy- and Veh-offspring (Supplementary Fig. S2b), thereby indicating the potential interaction between TET2 and PPAR α . In adulthood, we found no differences in the recruitment of TET2 between the mice treated with vehicle (Veh-mice) and those treated with Wy (Wy-mice) (Fig. 5c), which may explain that PPAR α -dependent DNA demethylation of *Fgf21* occurs only during the suckling period, a critical time window for DNA demethylation.”

2)-2 The fact that fgf21 demethylation seems to be limited to lactation, and is not induced during gestation, indeed suggests that the main inducing factor of epigenetic modulation is present in the mother milk. However, it was not

measured whether Wy14643 itself ends up in the mother milk, whether it is able to change the lipid composition and concentrations, or whether it might induce its effects through a synergistic reaction between Wy14643 and natural milk PPAR α ligands.

Our response

Following your comments and suggestions, by mass spectrometry, we tried to detect Wy14643 (Wy) in the offspring gastric contents at D16, indicating that Wy was transferred to the offspring via the breast milk. We could not determine Wy concentration in the milk. We included the data in the revised manuscript as **Supplementary Figure S1**. We also examined the milk lipid composition in the offspring gastric contents by gas chromatography and found no significant differences between Veh-offspring and Wy-offspring at D16. Taken together, these observations suggest that Wy, when transferred to offspring via the breast milk, would facilitate PPAR α -dependent DNA demethylation of *Fgf21*. We included the data in the revised manuscript as **Supplementary Table S1** and added the following sentences in **Result** section (at pages 8-9 on lines 144-151).

“To clarify whether Wy was transferred to pups via the breast milk, we employed the offspring gastric contents at D16, which mainly consisted of the milk derived from dams and applied them by mass spectrometry [liquid chromatography/tandem mass spectrometry: (LC/MSMS)]. As shown in Supplementary Figure S1, LC/MSMS detected same precursor [mass-to-charge ratio (m/z), 324.06] and product peaks (m/z, 306.04) in both standard and milk samples of Wy-offspring, indicating the presence of Wy in the breast milk of dams (Supplementary Fig. S1). In addition, to examine whether Wy altered milk lipid composition, we performed lipid composition analysis of milk using the offspring gastric contents by gas chromatography (GC). As shown in Supplementary Table S1, GC showed no significant difference in lipid composition of milk between Wy- and Veh-offspring, suggesting that Wy administration to dams during lactation period did not affect milk lipid composition (Supplementary Table S1).”

2)-3 Furthermore, if it is the case that solely the presence of Wy14643 in mother milk is able to induce PPAR α -mediated epigenetic memory, it cannot be

*concluded that the PPAR α ligands that are naturally present in mother milk could induce the same effect to a similar level. Therefore, to be able to draw the conclusions that were drawn in the present paper, it is needed to show that natural milk lipids, in physiological concentrations, are able to induce epigenetic modulation of *fgf21*.*

Our response

Even in the absence of Wy14643, PPAR α -dependent DNA demethylation of *Fgf21* could be induced during the suckling period. In the original manuscript, we demonstrated PPAR α -dependent DNA demethylation of *Fgf21* induced in the liver of wild-type offspring during the suckling period (**Figure 2b, c**), suggesting that DNA demethylation of *Fgf21* is physiologically induced by natural milk lipids. Thus, we added the following sentences in **Results** section (at pages 10-11 on lines 192-194), “Taken together, these observations suggest that PPAR α -dependent DNA demethylation of *Fgf21* physiologically occurs during the suckling period, after which it persists into adulthood.”

2)-4 In addition, measuring the lipid composition and Wy14643 levels in the mother milk, or including an extra control group where Wy14643 is administered directly to pups during lactation instead of via the mother milk, could give more insight into whether the epigenetic modulation depends on milk-derived factors, or solely on Wy14643 administration. Furthermore, it could indicate whether the lactation period is indeed the critical window of epigenetic modulation, or whether this is only suggested because epigenetic modulation depends on mother milk-specific synergistic mechanisms.

Our response

Following your comment, as described above, we detected Wy14643 in milk from Wy-administered dams. We analyzed milk lipid composition using the gastric contents of offspring by gas chromatography and found no significant differences between Veh-offspring and Wy-offspring at D16. Moreover, we found that palmitic acid, oleic acid and linoleic acid, which have been proposed for natural endogenous ligands for PPAR α (*Proc Natl Acad Sci USA* **90**, 2160-2164, 1993, *Nutr J* **13**, e17, 2014), are the major components of milk lipids (**Supplementary Table S1**).

According to your suggestion, we tried to administer Wy ligand directly to the pups. For the dose optimization, several concentrations of Wy were injected to the pups at D2, however the pups injected with Wy did not survive and were dead in a few days after the injection.

Therefore, we employed an alternative way to address your concern. Since fatty acids in diet would influence the lipid component of breast milk of dams (*J Dairy Sci* **98**, 431–442, 2015), we administered fat-free diet (0% energy as fat,) and control diet (10% energy as fat) to dams during the late gestation and lactation period and analyzed DNA methylation of *Fgf21* in the liver of offspring. We included the data of milk lipid composition and DNA methylation analysis as **Supplementary Table S2** and **Supplementary Figure S3**, respectively in the revised manuscript. Accordingly, we added the following sentences in **Results** section (at pages 12-13 on lines 234-253), “Because fatty acids in diet influence the lipid component of breast milk of dams, we administered fat-free diet (0% energy as fat) and control diet (10% energy as fat) to dams during late gestation and lactation period and analyzed DNA methylation of *Fgf21* in the liver of offspring (Supplementary Fig. S3). The lipid composition analysis of milk using the gastric contents of offspring derived from fat-free diet-fed dams revealed that linoleic acid and α -linolenic acid levels were markedly reduced relative to offspring derived from control diet-fed dams; notable, that eicosapentaenoic acid (EPA) was not detected in the milk of fat-free diet-fed dams (Supplementary Table S2).

However, DNA methylation status of *Fgf21* of the offspring derived from fat-free diet-fed dams was mainly unchanged relative to that derived from control diet-fed dams both at D16 and 4W (Supplementary Figure S3). Because the ratios of palmitic acid, oleic acid, arachidonic acid (ARA), and docosahexaenoic acid (DHA), in milk, which are known to be ligands for PPAR α , were comparable between the offspring derived from fed fat-free and control diet dams, it is likely that they are responsible for physiological DNA demethylation of *Fgf21*. Moreover, as the physiological DNA demethylation of *Fgf21* was induced without EPA in the milk components, we speculated that EPA may not be related to the DNA demethylation (Supplementary Table S2).”

In addition, as described above, we obtained evidence by ChIP assays that TET2 may be related to reduction of DNA methylation via PPAR α only during the suckling period,

which indicates that the suckling period is the critical time window of epigenetic modulation.

Major point 3

There is no mechanistic framework on how PPAR α activation may influence the methylation status of genes. In their previous paper, the authors speculate “that the ligand-activated PPAR α recruits factors involved in DNA demethylation to the promoter regions of the fatty acid β -oxidation genes via the PPAR-responsive element, thus leading to the gene-specific DNA demethylation.” One would expect inclusion of some experimental evidence on the actual mechanism of PPAR α -mediated demethylation in the present paper.

Our response

As described above, we performed ChIP assays for TET 1-3 and suggested that TET2 plays a role in DNA demethylation of *Fgf21* at D16. On the other hand, the recruitment of DNA methyltransferases (DNMTs: DNMT3a and 3b) to the promoter region of *Fgf21* was comparable between Wy-offspring and Veh-offspring at D16, suggesting that DNMTs were not related to PPAR α -dependent DNA demethylation.

In the revised manuscript, we have included the data on ChIP analysis as **Figure 5** (newly assigned) with the statement on the molecular mechanism for PPAR α -dependent DNA demethylation of *Fgf21* in **Results** and **Discussion** sections (Please see our response to Major point 2).

Minor point 1

*In figure 2a, the BS region of *fgf21* is shown. As stated by the authors, two PPAR response elements (PPRE) were found near the TSS of the gene. However, the BS region of *fgf21* only includes the second PPRE (100 bp upstream of the TSS) that was found in silico, and no methylation status is measured for the first PPRE (1000 bp upstream of the TSS), meaning that the role of the first PPRE is not taken into account in the bisulfite analysis and further analyses. Please speculate on the potential consequences of excluding the first PPRE.*

Our response

Since the first PPRE (TGGCCTGTGGCCA) includes no CpG sites, we did not evaluate DNA methylation status for this region.

Minor point 2-1

In the text concerning figure 3, it could be stated more clearly that periods i and ii actually consist of two distinct experiments.

Our response

According to your suggestion, we corrected the sentences on **Figure 3** in **Results** section as follows; “In this study, we divided dams into two groups; one was treated with Wy only during the late gestation period (e14–18) [**Fig. 3a** (i)], and the other was treated with Wy only during the lactation period (D2–16) [**Fig. 3a** (ii)]. We found that reductions in DNA methylation of *Fgf21* were significantly enhanced in Wy-offspring relative to Veh-offspring during the lactation but not the late gestation period.”

Minor point 2-2

*Furthermore, no explanation is given as to why Wy14643 might not be able to induce epigenetic modulations of *fgf21* during adulthood.*

Our response

As described above, we performed ChIP assays for TET 1-3 and suggested that TET2 plays a role for DNA demethylation of *Fgf21* at D16 (**Figure 5b**, newly assigned). At 6W in adulthood, the recruitment of TET2 to the promoter region of *Fgf21* was roughly equivalent between the mice treated with vehicle (Veh-mice) and those treated with Wy (Wy-mice) (**Figure 5c** in the revised manuscript), which may explain that PPAR α -dependent DNA demethylation of *Fgf21* occurs only during the suckling period, a critical time window for DNA demethylation. In the revised manuscript, we have added the statement as to why Wy14643 might not be able to induce epigenetic modulations of *Fgf21* during adulthood in **Results** and **Discussion** sections (Please see our response to Major point 2).

Minor point 3

*In figure 4, large differences in mRNA expression of *fgf21* and serum FGF21 levels can be seen between D2 and D16, although both periods fall in the period of lactation that was defined as the critical window of epigenetic *fgf21* modulation by the authors. Please provide a potential explanations for the decreased expression and serum levels at D16 compared to D2.*

Our response

Developmental FGF21 expression was investigated previously (*Cell Metab.* **11**, 206-212, 2010). Although serum FGF21 levels in fetus are very low upon birth, they increase rapidly to the highest levels within 2 days after birth and remained high in 6-day-old mice and declined thereafter. At 3 weeks, the time at which weaning was completed, serum FGF21 levels are similar to those in adults. A similar pattern is observed for hepatic FGF21 gene expression.

The rapid increase in FGF21 expression in neonates upon birth is attributed to the initiation of suckling, which facilitates high serum levels of free fatty acids derived from lipid content of milk. Circulating free fatty acids levels are markedly increased 16 hours after birth with a peak at D2 and decreased thereafter toward weaning (D16-D21) (*Cell Metab.* **11**, 206-212, 2010). Indeed, we also confirmed that plasma free fatty acids levels at D2 were higher than those at D16 in our previous report (*Diabetes* **64**, 775-784, 2015). These changes in plasma free fatty acids levels in postnatal period may be attributed to the development of white adipose tissue. Since hepatic FGF21 gene expression during development is under the control of fatty acids that reach the liver (*Cell Metab.* **11**, 206-212, 2010), it is rational that *Fgf21* mRNA levels and serum FGF21 concentrations at D2 are higher than those at D16. This could be a physiological phenomenon, which is not affected by the DNA methylation status of *Fgf21*.

Minor point 4

The legend of figure 4d mentions 3-8 animals per group. However, the right panel (correlation) of figure 4d shows only 3-4 mice per group.

Minor point 5

Performing a correlation analysis for n=7-8 is meaningless

Our response

In the right panel of **Figure 6d** (newly assigned), we examined if there is a correlation between the degree of DNA methylation (% DNA methylation) and the induction of gene expression. Following your comments, we performed additional experiments with 11-16 mice and confirmed a correlation between the degree of DNA methylation (% DNA methylation) and the induction of gene expression in the revised manuscript (**Figure 6c, d, e**, newly assigned). Based upon our bioinformatician's suggestion (Dr.

Takako Takai-Igarashi), we also showed 95% confidence interval (CI) for each correlation to validate statistical relevance of sample size.

Minor point 6

Figure 4c and 4d: Can the authors explain how it is possible that fgf21 plasma levels are much more highly induced upon Wy injection as opposed to fgf21 mRNA in liver. It suggests that a significant portion of plasma fgf21 is not derived from liver.

Our response

It has been reported that serum FGF21 is exclusively derived from the liver upon Wy administration (*Sci Rep* **6**, e30484, 2016). As described in the original manuscript, we evaluated *Fgf21* mRNA levels 1 hour after Wy injection, whereas serum FGF21 concentrations were measured 3 hours after Wy injection, which may result in the difference between *Fgf 21* mRNA levels and serum FGF21 concentrations. In addition, increases in *Fgf 21* mRNA and serum FGF21 concentrations upon Wy administration, which are similar to those achieved in this study, were reported previously (*Sci Rep* **6**, e30484, 2016; *Physiol Res* **63**, 483-490, 2014). Therefore, the induction of serum FGF21 concentrations and *Fgf 21* mRNA levels upon Wy administration is physiologically relevant.

Minor point 7

At figure 5b, significant differences that are indicated for the body weight between Veh-HFD and Wy-HFD groups in week 11 to 14 seem very small (the graphs indicates differences around 2-3 grams). From our experience, it seems doubtful that these differences can be significant in an experiment conducted with 11 mice per group. Please verify statistical significance.

Our response

According to the suggestion from our bioinformatician, Dr. Takako Takai-Igarashi, we reevaluated the statistical analysis for body weight changes (**Figure 7b** newly assigned which was originally **Figure 5b**). To verify statistical significance, we compared the differences in body weight between Veh-HDF and Wy-HFD at 0 to 14 weeks. Comparison of body weight difference was evaluated by two-way ANOVA (between-group, within-time, and interaction of time and group). Furthermore,

comparison between groups at each week of age was evaluated by multiple *t*-test with Bonferroni correction. We added the statement on the statistical analysis in **Methods** section. [REDACTED]

General recommendation: The conclusions that are drawn in the present paper seem heavily based on correlations and speculations. The authors should indicate whether fgf21 epigenetic memory could be induced specifically during the lactation period in natural conditions or whether only the administration of Wy14643 is able to induce epigenetic modulation, and provide more substantial evidence that the induction of fgf21 demethylation is indeed largely responsible for the attenuation of diet-induced obesity, or otherwise tone down the conclusions that are drawn.

Our response

As described above, we have demonstrated that DNA demethylation of FGF21 gene is induced and DNA methylation status established during the suckling period is maintained to adulthood in wild-type mice, which are under physiological conditions. In addition, using FGF21-deficient (KO) mice, we provide evidence that increased FGF21 as a result of DNA demethylation is at least in part responsible for the metabolic phenotype of Wy-offspring in adulthood. Nonetheless, since it remains possible that other epigenetic memory genes such as *Ucp3* could contribute to the attenuation of diet-induced obesity in adulthood, we have changed the title of the manuscript to **‘Epigenetic modulation of fibroblast growth factor 21 in the perinatal mouse liver is associated with the attenuation of diet-induced obesity in adulthood.’**

To Reviewer #3

Thank you very much for your valuable comments and suggestions. According to your comments and suggestions, we have revised the manuscript as follows.

General comments

The authors report an examination of DNA methylation and expression dynamics of FGF21 in peri and postnatal mouse liver and correlate these findings to attenuation of diet-induced obesity in adulthood. The study is interesting and is overall well written. The data measures and analysis appear well performed and are appropriate. It suggests DNA-methylation dynamics associated with Ppara binding near the FGF21 promoter early in life can program FGF21 expression stably and over the long term. The data are of interest but still fall short of convincing the reader on all aspects particularly regarding technical methodology and mechanism. If the authors can address key points below, the ideas and experiments presented here would be appropriate for this journal.

Major concern 1

Specifically, it is not entirely clear that this is FGF21 mediated. If FGF21 mediated, the effects of the intervention should be abolished in either liver FGF21 knockout animals, or through chronic immunoneutralization of FGF21 in control and reprogrammed animals. Can these experiments be done?

Our response

Following your comment, we obtained FGF21-deficient (KO) (strain: C57BL/6NJcl, global knockout) mice, which are kindly provided by Prof. Morichika Konishi in Kobe Pharmaceutical University, Hyogo, Japan (*Endocrinology* **150**, 4625-4633, 2009) and induced DNA demethylation of all the PPAR α target genes except *Fgf21* by maternal administration of Wy. We examined the metabolic phenotype of the offspring derived from FGF21-KO dams treated with Wy and Veh during 10 weeks of HFD feeding (4–14W). We included the data as **Figure 8** (newly assigned in the revision) and added the following sentences in **Results** section (at pages 20-21 on lines 410-424),

“We obtained FGF21-deficient (KO) mice and induced DNA demethylation of all the PPAR α target genes except *Fgf21* by maternal administration of Wy (Fig. 8a). We

examined the metabolic phenotype of the offspring derived from FGF21-KO dams treated with Wy and Veh during 10 weeks of HFD feeding (4–14W) (Veh-FGF21-KO and Wy-FGF21-KO, respectively) (Fig. 8a). As shown in Figure 8b, we found no significant difference in body weight between Veh-FGF21-KO and Wy-FGF21-KO. The weights of iWAT, eWAT, and BAT were roughly comparable between the Veh-FGF21-KO and Wy-FGF21-KO at 14W (Fig. 8c). Serum FGF21 was not detected in both groups, proving systemic FGF21 deficiency (Fig. 8d). Histologically, adipocyte cell size in eWAT appeared to be comparable between the Veh-FGF21-KO and Wy-FGF21-KO groups at 14W (Fig. 8e). Consistently, the mRNA expression of *Egr1*, *c-fos*, *Hsl* and *Atgl* showed no significant difference between the two groups (Fig. 8f). These observations are consistent with the notion that *Fgf21* plays a major role in the metabolic phenotypes of Wy-HFD (Fig. 7).”

We also added the following sentences in **Discussion** section (at page 29 on lines 595-599),

“However, since the metabolic phenotypes observed in Wy-HFD were cancelled in Wy-FGF21-KO mice, we speculated that *Fgf21* through DNA demethylation induced during the suckling period may be at least in part associated with the attenuation of diet-induced obesity in adulthood.”

Major concern 2-1

Second, the epigenetic mechanism is not entirely convincing. In the end, the authors are proposing that an ~15 % change in DNA-methylation is responsible for the changes in FGF21 expression. How do the authors proposed this results in altered gene regulation.

Our response

[REDACTED].

The correlation between DNA methylation status and gene expression has also been reported previously. For instance, about 3 % change in DNA methylation of corticotropin releasing hormone gene resulted in 3-fold change in gene expression in a trophoblastic cell line, which supports the idea that subtle differences in DNA methylation would induce substantial differences in gene expression and is compatible to our data (*Int J Endocrinol.* **2015**:861302, 2015). Taken together with our preliminary data and findings in the previous reports, in the revised manuscript, we added the following sentences in **Discussion** section (at page 25 on lines 499-501), “Indeed, it has been known that subtle but significant differences in DNA methylation would induce substantial differences in gene expression”

Major concern 2-2

Which CpG sites are involved?

Our response

We evaluated DNA methylation ratios of each CpG site in the promoter region of *Fgf21* and correlation between the DNA methylation ratios and the induction of *Fgf21* expression upon the single Wy injection. We have included the data as **Supplementary Figure S6** (newly assigned in the revision) and added the following sentences in **Results** section (at page 17 on lines 338-343),

“We evaluated DNA methylation ratios of each CpG site in the promoter region of *Fgf21* at 14W (Supplementary Figure S6a) and found that CpG sites located downstream of the transcription start site (TSS) (the CpG site number: #10-21) are sensitive to PPAR α -dependent DNA demethylation. The CpG sites but not those located upstream of the TSS were correlated to the induction of *Fgf21* expression upon the single Wy injection (Supplementary Figure S6b, c).”

We have also added the following sentences in **Discussion** section (at page 24 on lines 472-491),

“DNA methylation of the promoter region is a repressive epigenetic mark that down-regulates gene expression. DNA demethylation of *Fgf21* initiated at the promoter region, suggesting derepression of the gene expression upon the onset of breast feeding. DNA methylation ratios of CpG sites downstream of TSS, in gene body, were significantly different between Veh- and Wy-offspring, suggesting that enhanced DNA demethylation via ligand-activated PPAR α could specifically occur at these CpG sites (Supplementary Fig. S6a, c). Since DNA methylation ratios of these CpG sites were negatively correlated to the induction of gene expression, DNA hypomethylation status at these CpG sites could determine the magnitude of the gene expression response to environmental cues (Supplementary Fig. S6c : right). So far, possible functions of gene body or intragenic DNA methylation has not yet been fully elucidated. It was reported that intragenic DNA methylation in mammalian cells initiates formation of a chromatin structure that reduces the efficacy of Pol II elongation, thereby repressing the gene expression, which is compatible to our data. On the other hand, it has been recently reported that alternative TSSs could be located at CpG sites in gene body, which are extensively methylated. Therefore, DNA demethylation of those CpG sites could derepress the alternative TSSs and enhance the gene expression, which can be another explanation of our findings.”

Major concern 2-3

How do they regulate FGF21 expression? How is transcription and transcription factor binding so reproducibly affected, by these modest changes?

Our response

We performed ChIP assays on the promoter region of *Fgf21* to evaluate the recruitment of RNA polymerase II (Pol II), a general transcription factor at D16 under Wy administration via breast feeding and at 14W upon a single Wy injection. In the revised manuscript, we have included the data as **Figure 6g and 6h** (newly assigned) and added the following sentences in **Results** section (at page 17 on lines 331-337),

“ChIP assays revealed that RNA polymerase II (Pol II) was recruited abundantly to the promoter region of *Fgf21* of Wy-offspring relative to Veh-offspring at D16, suggesting active transcription of *Fgf21* in Wy-offspring relative to Veh-offspring. On the other hand, the recruitment of PPAR α was roughly equivalent between Wy- and Veh-offspring (Fig. 6g). Upon a single Wy injection at 14W, Pol II was recruited

abundantly to the promoter region of *Fgf21* of Wy-offspring relative to Veh-offspring (Fig.6h). This data suggest that a modest difference in DNA methylation status can affect transcriptional activity.”

Major concern 2-4

The DNA methylation correlation to FGF expression on the mouse to mouse basis is nice. Are there specific CpG's where DNA methylation changes correlate poorly to the severity of or FGF expression? Are there others where this correlation is perfect?

Our response

As described above, we evaluated DNA methylation correlation to *Fgf21* expression of each CpG site in *Fgf21* and have included the new data as **Supplementary Figure S6** (Please see our response to Major concern 2-2).

Major concern 2-5

Is there cellular heterogeneity in expression of FGF21 in the liver, does this change with the treatments (ie. the bisulfite shows that some cells have a fully methylated sequence and others fully unmethylated).

Our response

We agree with your comment that there may be cellular heterogeneity in DNA methylation status. In the revised manuscript, we added the statement on cellular heterogeneity in DNA methylation status in **Discussion** section (at page 25 on lines 492-496) as follows;

“Bisulfite sequencing analysis also revealed that DNA methylation status of *Fgf21* was not monotonous, suggesting cellular heterogeneity in DNA methylation status. Furthermore, because the liver consists of various types of cells such as hepatocytes, stellate cells, Kupffer cells and sinusoidal endothelial cells, it may be ideal to collect hepatocytes for bisulfite sequencing.”

Major concern 3

The phenotype is very mild, which is fine. That said, the chance that this is simple variation between individuals is thus enhanced. The third important concern is therefore that there is little description of the exact mouse husbandry

and animal choice that was performed. Metabolic changes can be observed between mice born of different litter sizes, or born to different fathers, or born to virgin vs multiparous mothers. How many of these variables were controlled and how. In the end, the key features of the study must be repeated with maternally matched animals ie. either virgin or multiparous mothers, comparing only equally sized litters. Were all the litter sizes equal? Overall the methods section suggests that the experiment was done in one large cohort... If so this needs to be repeated.

Our response

Following your comment, we added the statement on husbandry related controls in **Method** section (at page 31 on lines 626-629),

“In each experiment, we purchased pregnant female C57BL6 mice, which were primiparous, from CLEA Japan (Tokyo, Japan) at gestational day 13. After delivery, litter size was adjusted to 5~6 pups (all male) per dam to avoid metabolic drifts due to nutrient availability during lactation.”

(at page 31 on lines 637-639), “We repeated the experiments with the same protocol six times with reproducible results and showed the representative data.”

[REDACTED]

Major concern 4

It is not clear to me the mechanism by which Wy is exerting its effect, direct (ie. passing through the milk) or indirectly (affecting the mothers metabolism and thus the composition of the milk). This should be addressed. Can the authors measure Wy in the milk? Has it been ruled out that the Wy compound is working directly in the offspring ie. getting into the milk? Alternatively, can the authors identify a milk component that can mimic the effect?

Our response

Following your comments and suggestions, by mass spectrometry, we tried to detect Wy14643 (Wy) in the offspring gastric contents at D16, indicating that Wy was transferred to the offspring via the breast milk. We could not determine Wy concentration in the milk. We included the data in the revised manuscript as **Supplementary Figure S1**. We also examined the milk lipid composition in the

offspring gastric contents by gas chromatography and found no significant differences between Veh-offspring and Wy-offspring at D16. Taken together, these observations suggest that Wy, when transferred to offspring via the breast milk, would facilitate PPAR α -dependent DNA demethylation of *Fgf21*. We included the data in the revised manuscript as **Supplementary Table S1** and added the following sentences in **Results** section (at pages 8-9 on lines 144-151),

“To clarify whether Wy was transferred to pups via the breast milk, we employed the offspring gastric contents at D16, which mainly consisted of the milk derived from dams and applied them by mass spectrometry [liquid chromatography/tandem mass spectrometry: (LC/MSMS)]. As shown in Supplementary Figure S1, LC/MSMS detected same precursor [mass-to-charge ratio (m/z), 324.06] and product peaks (m/z, 306.04) in both standard and milk samples of Wy-offspring, indicating the presence of Wy in the breast milk of dams (Supplementary Fig. S1). In addition, to examine whether Wy altered milk lipid composition, we performed lipid composition analysis of milk using the offspring gastric contents by gas chromatography (GC). As shown in Supplementary Table S1, GC showed no significant difference in lipid composition of milk between Wy- and Veh-offspring, suggesting that Wy administration to dams during lactation period did not affect milk lipid composition (Supplementary Table S1).”

Followed by the reviewer’s suggestion, we tried to identify a natural milk component responsible for PPAR α -dependent DNA demethylation of *Fgf21*.

The lipid composition analysis of milk using the gastric contents of offspring revealed that palmitic acid, oleic acid and linoleic acid, which have been proposed as natural endogenous ligands for PPAR α (*Proc Natl Acad Sci USA* **90**, 2160-2164, 1993, *Nutr J* **13**, e17, 2014), are the major components of milk lipids (**Supplementary Table S1**).

Since fatty acids in diet would influence the lipid component of breast milk of dams (*J Dairy Sci* **98**, 431–442, 2015), we administered fat-free diet (0% energy as fat,) and control diet (10% energy as fat) to dams during the late gestation and lactation period and analyzed DNA methylation of *Fgf21* in the liver of offspring. We included the data of milk lipid composition and DNA methylation analysis as **Supplementary Table S2** and **Supplementary Figure S3**, respectively in the revised manuscript. Accordingly, we added the following sentences in **Results** section (at pages 12-13 on lines 234-253), “Because fatty acids in diet influence the lipid component of breast milk of dams, we

administered fat-free diet (0% energy as fat) and control diet (10% energy as fat) to dams during late gestation and lactation period and analyzed DNA methylation of *Fgf21* in the liver of offspring (Supplementary Figure S3). The lipid composition analysis of milk using the gastric contents of offspring derived from fat-free diet-fed dams revealed that linoleic acid and α -linolenic acid levels were markedly reduced relative to offspring derived from control diet-fed dams; notable, that eicosapentaenoic acid (EPA) was not detected in the milk of fat-free diet-fed dams (Supplementary Table S2).

However, DNA methylation status of *Fgf21* of the offspring derived from fat-free diet-fed dams was mainly unchanged relative to that derived from control diet-fed dams both at D16 and 4W (Supplementary Figure S3). Because the ratios of palmitic acid, oleic acid, arachidonic acid (ARA), and docosahexaenoic acid (DHA), in milk, which are known to be ligands for PPAR α , were comparable between the offspring derived from fed fat-free and control diet dams, it is likely that they are responsible for physiological DNA demethylation of *Fgf21*. Moreover, as the physiological DNA demethylation of *Fgf21* was induced without EPA in the milk components, we speculated that EPA may not be related to the DNA demethylation (Supplementary Table S2).”

Accordingly, we included the following sentences in **Discussion** section (at pages 23-24 on lines 465-471),

“It is of great physiological significance to identify a milk lipid component that acts as a PPAR α agonist and might thus mediate DNA demethylation of *Fgf21*. Through milk lipid composition analysis of veh- and Wy-treated and fat-free diet fed dams, we speculated that the fatty acids such as palmitic acid, oleic acid, ARA and DHA, forming a coordination ligand complex for PPAR α , would mediate PPAR α -dependent DNA demethylation of *Fgf21* (Supplementary Tables. S1 and S2).”

Additional concern 1

Figure 1 – please show all the data. A volcano plot should be shown for each D6 and 14W timepoints to give an indication of the spread and noise of the data. A correlation plot showing the differences at 14W (y-axis) vs. D6 (x-axis) to show the correlation between the datasets should be shown.

Significantly hyper and hypo methylated alleles should be highlighted on the plot. A heatmap would give important information of the noise and variation of the data for instance.

Our response

In the promoter array employed in MIAMI analysis, only one or two probes are mounted for some genes, for which p-values for a volcano plot are not calculated for in a single MIAMI analysis. Alternatively, we showed entire MIAMI data comparing Veh-offspring and Wy-offspring at D16 and 14W (**Figure 1b and c**, newly assigned), as we reported in the previous study (*Diabetes* **64**, 775-784, 2015). Following your suggestion, we added a correlation plot showing the differences at D16 (X-axis) vs. 14W (Y-axis), suggesting that DNA methylation status at D16 is weakly but significantly correlated to that at 14W (**Figure 1d**, newly assigned). Genes, which are hypomethylated both at D16 and 14W, are highlighted as blue circles.

Additional concern 2

Throughout the manuscript the authors talk about DNA demethylation. While not entirely incorrect, the most common use of the term in the literature implies an active process. Since the authors provide no evidence of active DNA demethylation, they should refer throughout to reductions or losses in DNA methylation.

Our response

In the revision, we obtained evidence by ChIP assays that TET2 is related to reduction of DNA methylation via PPAR α , suggesting active DNA demethylation (*Nat Rev Genet*, **18**, 517-534, 2017). Therefore, we used the term ‘DNA demethylation’. However, followed by the reviewer’s suggestion, we reviewed the term ‘DNA demethylation’ throughout the manuscript in this revision and corrected it to ‘reductions in DNA methylation’, accordingly.

Additional concern 3

The quantification of adipocyte size appear not to be normalized properly. The area under the curve should be equal adding up to 100% in both curves. This doesn’t appear to have been done in Figure 5 and Supp Fig 5. This analysis

should be redone and the median value and interquartile range highlighted. Statistical analysis based a number of mice should also be applied.

Our response

Following the reviewer's suggestion, we reevaluated the quantification of adipocyte size and performed statistical analysis based on a number of mice. In this revision, we put the relative frequencies on the vertical axis, so that the frequencies should add up to 100%. We also highlighted the median value and interquartile range in the histogram both in **Figure 7e** and **Supplementary Figure 8e** (newly assigned in the revised manuscript).

Additional concern 4

Figure 3. I realize it is substantial work but I'm not sure I agree that the DNA methylation is unchangeable in the adult. The Wy experiment in 3b should be repeated with a longer intervention. There is a clear (non-significant change in the figure shown). Do these authors really believe this couldn't be altered in adulthood by Wy?

Our response

[REDACTED] Thus, we believe it is unlikely that DNA methylation status of *Fgf21* could be altered by Wy administration in adulthood.

Additional concern 5

Similarly, the authors should perform more replicates of the data in Supp Fig b-d. There are trends in H3K27me3 and Ppara ChIP that could be very important if they are real. These data should be done with many replicates and included in the main Figure. I can very much imagine decreased K27me3 and enhanced Ppara binding at the locus even constitutively. It is important not to hide this in the supplements or state no change, just because the error is substantial. Overall the authors appear to have done very precise work with very tight error where they want to see a difference. They should strive for the same here.

Our response

Following your suggestion, we repeated ChIP assays for transcription factors such as PPAR α and histone marks such as H3K4me3, H3K27Ac, H3K27me3 and H3K9me2 both at D16 and 14W. Thus, based on the data by repeated experiments, we have concluded that there was no significant difference in the recruitment of PPAR α or in the levels of active and repressive histone marks in the promoter region of *Fgf21* between Wy- and Veh-offspring both at D16 and 14W. We included the data as main Figures in the revised manuscript (**Figure 4b, c**). [REDACTED]. In addition, we added the sentences regarding the recruitment of PPAR α in **Results** section (at page 14 on lines 272-274),

“On the other hand, the recruitment of PPAR α was roughly equivalent between Wy- and Veh-offspring both at D16 and 14W (Fig. 4d).”

Minor concern 1

One additional suggestion, that perhaps is for the future for the authors. Mice compensate for high-fat diet extremely well if HFD is started at a young age. Here the authors have started HFD at the youngest possible age when the mice are only half grown. The authors may find more dramatic metabolic shifts if HFD is first applied after 10 weeks of age.

Our response

We really appreciate your suggestion. We have added the following sentences in **Discussion** section (at page 27 on lines 538-542),

“Even though statistically significant, the differences in body and eWAT weight between Wy-HFD and Veh-HFD were relatively small. We speculated that alternative protocols for HFD feeding, for example HFD feeding starts after 10W, might enhance the difference.”

Minor concern 2

The methods section could use more details. There is a 'mm' typo in the histology section. How many sections and how far apart were the sections counted from each mouse?

Our response

Following your suggestion, we corrected the errors and described the details of sections in **Methods** section.

Minor concern 3

Given the time-frame of best induction, I'm not sure peri-natal is the best term. Perhaps post-natal?

Our response

We used 'peri-natal' period as pregnancy and lactation period. Following your suggestion, we corrected the word.

REVIEWERS' COMMENTS:

Reviewer #1 (Remarks to the Author):

This is a very nice study showing the epigenetic regulation of FGF21, an important metabolic hormone secreted from the liver. The revised version has addressed all the concerns I raised in the first version.

Reviewer #2 (Remarks to the Author):

The authors have substantially improved the manuscript via several additional experiments. However, without providing clear evidence on how provision of Wy14643 to the dams affects epigenetic regulation in the offspring, the paper leaves a large conceptual gap. The paper is incomplete without providing clarity on the issue of how provision of Wy14643 to the dams causes epigenetic modulation in the offspring. It must be through the milk but no changes in either lipid content or Wy14643 content in the milk were detected. Without this information, it is unclear what we are actually looking at.

Specifically, no changes in lipid content in response to Wy14643 were reported. Also, Wy14643 could not be detected in the milk, although the description of the data in the results and rebuttal is very confusing. For example, the paper states in line 150 "indicating the presence of Wy in the breast milk of dams (Supplementary Fig. S1)". Presence should be absence. In the rebuttal it is stated that "We could not determine Wy concentration in the milk. ". Nevertheless, a few lines down it says: "Taken together, these observations suggest that Wy, when transferred to offspring via the breast milk, would facilitate PPAR α -dependent DNA demethylation of Fgf21. The authors are dodging this key point via a poor and inconsistent description.

Reviewer #3 (Remarks to the Author):

The authors should be commended for hard work during the revisions. Overall, the work in the manuscript is impressive and appears of very high quality. This is important as much of the field of intergenerational and developmental reprogramming suffers from relatively low quality experiments and data. Here, the authors stand tall and should be applauded. They have done an equally strong attempt to address all the comments I had. I don't see any reason to add any substantial experiments. My few remaining concerns should be easily addressed:

Old Additional Concern 5:

I would still argue that the changes in the repressive histone marks, though still non-significant, are still substantial. They must now be even closer to significant :-). Rather than ask the authors to spend more time repeating experiments, however, I would request that they acknowledge in the text that "The current report highlights loss of DNA methylation at the locus to be a potential mechanistic driver of the effects. Given the targeted nature of these assays used, and in light of the trends observed in several histone marks at the targeted loci, it should be noted that the data do not completely rule out contribution of related epigenetic silencing / desilencing mechanisms."

English editor:

Overall the manuscript is well and carefully written. A native English language editor could still be beneficial in reducing a few instances where word choice is still a little ambiguous. Well done to the authors.

To Reviewer #2

Thank you very much for your valuable comments and suggestions. According to your comments and suggestions, we have revised the manuscript as follows.

General comments

The authors have substantially improved the manuscript via several additional experiments. However, without providing clear evidence on how provision of Wy14643 to the dams affects epigenetic regulation in the offspring, the paper leaves a large conceptual gap. The paper is incomplete without providing clarity on the issue of how provision of Wy14643 to the dams causes epigenetic modulation in the offspring. It must be through the milk but no changes in either lipid content or Wy14643 content in the milk were detected. Without this information, it is unclear what we are actually looking at.

Major point

Specifically, no changes in lipid content in response to Wy14643 were reported. Also, Wy14643 could not be detected in the milk, although the description of the data in the results and rebuttal is very confusing. For example, the paper states in line 150 “indicating the presence of Wy in the breast milk of dams (Supplementary Fig. S1).”. Presence should be absence. In the rebuttal it is stated that “We could not determine Wy concentration in the milk. “. Nevertheless, a few lines down it says: “Taken together, these observations suggest that Wy, when transferred to offspring via the breast milk, would facilitate PPAR α -dependent DNA demethylation of Fgf21. The authors are dodging this key point via a poor and inconsistent description.

Our response

We are sorry for confusing and misleading you due to our ambiguous explanation and obscure expression.

As we demonstrated in Supplementary Figure 1, LC/MS-MS detected Wy in the milk samples of Wy-offspring (derived from Wy-treated dams) qualitatively. Therefore, we are confident that Wy is present in the milk samples of Wy-offspring. However, we could not determine the exact Wy concentration in the milk samples in a quantitative way, which may be a limitation of LC/MS-MS. To avoid misleading the readers, we

corrected the statements in **Result** section regarding detection of Wy in the milk samples of Wy-offspring as follows (Page 8, lines 141-149, in re-revised manuscript);

“To clarify whether Wy was transferred to pups via the breast milk, we analyzed gastric contents, which mainly consisted of the milk derived from dams, in offspring at D16 using mass spectrometry (liquid chromatography/tandem mass spectrometry [LC/MS-MS]). As shown in Supplementary Figure 1, LC/MS-MS detected the same precursor (mass-to-charge ratio [m/z], 324.06) and product peaks (m/z, 306.04) in both a standard sample consisting of purified Wy and milk samples from Wy-offspring (derived from Wy-treated dams), suggesting that Wy is present in the breast milk of dams (Supplementary Fig. 1).”

To Reviewer #3

Thank you very much for your valuable comments and suggestions. According to your comments and suggestions, we have revised the manuscript as follows.

General comments

The authors should be commended for hard work during the revisions. Overall, the work in the manuscript is impressive and appears of very high quality. This is important as much of the field of intergenerational and developmental reprogramming suffers from relatively low quality experiments and data. Here, the authors stand tall and should be applauded. They have done an equally strong attempt to address all the comments I had. I don't see any reason to add any substantial experiments. My few remaining concerns should be easily addressed:

Major point

*I would still argue that the changes in the repressive histone marks, though still non-significant, are still substantial. They must now be even closer to significant :-)
Rather than ask the authors to spend more time repeating experiments, however, I would request that they acknowledge in the text that "The current report highlights loss of DNA methylation at the locus to be a potential mechanistic driver of the effects. Given the targeted nature of these assays used, and in light of the trends observed in several histone marks at the targeted loci, it should be noted that the data do not*

completely rule out contribution of related epigenetic silencing / desilencing mechanisms."

Our response

Following your comment, we corrected and added the statement regarding histone modification in **Discussion** section (at page 24 on lines 471-478),

“In this study, we did not observe a significant difference of histone modification at 14W, although active marks were more enriched in Wy-offspring than in Veh-offspring on D16. However, given the targeted nature of ChIP assays, and in light of the trends observed in several histone marks at the targeted loci, it should be noted that the data do not completely rule out the contribution of histone-related gene silencing/desilencing mechanisms to the epigenetic memory.”

We thank you and the reviewers again for your valuable comments and suggestions. We believe that our manuscript has been improved and hope that it is now suitable for publication in *Nature Communications*. We would greatly appreciate your kind arrangement and consideration of our manuscript.